# Sirt6 deficiency exacerbates podocyte injury and proteinuria through targeting Notch signaling

Min Liu[1], Kaili Liang[1], Junhui Zhen[2], Meng Zhou[1], Xiaojie Wang[1], Ziying Wang[1], Xinbing Wei[1], Yan Zhang[1], Yu Sun[1], Zhuanli Zhou[3], Hua Su[3], Chun Zhang (ID) [3], Ningjun Li[4], Chengjiang Gao[5], Jun Peng[6] & Fan Yi[1]

Podocyte injury is a major determinant of proteinuric kidney disease and the identification of potential therapeutic targets for preventing podocyte injury has clinical importance. Here, we show that histone deacetylase Sirt6 protects against podocyte injury through epigenetic regulation of Notch signaling. Sirt6 is downregulated in renal biopsies from patients with podocytopathies and its expression correlates with glomerular filtration rate. Podocyte-specific deletion of Sirt6 exacerbates podocyte injury and proteinuria in two independent mouse models, diabetic nephropathy, and adriamycin-induced nephropathy. Sirt6 has pleiotropic protective actions in podocytes, including anti-inflammatory and anti-apoptotic effects, is involved in actin cytoskeleton maintenance and promotes autophagy. Sirt6 also reduces urokinase plasminogen activator receptor expression, which is a key factor for podocyte foot process effacement and proteinuria. Mechanistically, Sirt6 inhibits Notch1 and Notch4 transcription by deacetylating histone H3K9. We propose Sirt6 as a potential therapeutic target for the treatment of proteinuric kidney disease.

[1] Department of Pharmacology, Shandong University School of Medicine, Jinan 250012, China. [2] Department of Pathology, Shandong University School of Medicine, Jinan 250012, China. [3] Department of Nephrology, Union Hospital, Tongji Medical College, Huazhong University of Science and Technology, Wuhan 430022, China. [4] Department of Pharmacology and Toxicology, Medical College of Virginia, Virginia Commonwealth University, Richmond, VA 23298, USA. [5] Department of Immunology, Shandong University School of Medicine, Jinan 250012, China. [6] Department of Hematology, Qilu Hospital, Shandong University, Jinan 250012, China. Correspondence and requests for materials should be addressed to F.Y. (email: fanyi@sdu.edu.cn)

P odocytes are highly specialized, terminally differentiated epithelial cells that are integral components of the renal glomerular filtration barrier, which are vulnerable to a variety of injuries and as a result, they undergo a series of changes ranging from hypertrophy, detachment, autophagy to apoptosis[1]. The genetic or acquired impairment of podocytes leads to proteinuria and contributes to the development and progression of proteinuric kidney disease. As podocytes have limited ability to repair and regenerate, the extent of podocyte injury is considered as a major prognostic determinant in end-stage renal disease[2]. There is mounting evidence of podocyte injury in a variety of other renal diseases such as focal segmental glomerulosclerosis

**Fig. 1** Sirt6 is significantly reduced in podocytes from mice with diabetic nephropathy or ADR nephropathy. **a** Representative western blot gel documents and summarized data showing the relative protein levels of SIRTs in the kidney from STZ-induced diabetic mice and ADR-treated mice. *$P < 0.05$ vs. control mice. ($n = 8$.) **b** Representative western blot gel documents and summarized data showing the relative protein levels of Sirt6 in the kidney from diabetic *db/db* mice and their genetic control *db/* + mice. *$P < 0.05$ vs. *db/+* mice. ($n = 8$.) **c** Representative confocal microscopic images showing the expression of Sirt6 in podocytes from STZ-induced diabetic mice and ADR-treated mice, synaptopodin were used as podocyte marker. The *arrows* indicate representative podocytes. *Scale bar*, 25 μm. **d** Representative western blot gel documents and summarized data showing the relative protein levels of Sirt6 in podocytes treated with high glucose (HG, final concentration 20 or 40 mmol/l in medium) for 24 h. *$P < 0.05$ vs. control. ($n = 6$.) **e** Representative western blot gel documents and summarized data showing the relative protein levels of Sirt6 in podocytes with advanced glycation end-product (AGE, 50–200 μg/ml) for 24 h. *$P < 0.05$ vs. control. ($n = 6$.) **f** Representative western blot gel documents and summarized data showing the relative protein levels of Sirt6 in podocytes with ADR (ADR, 0.4 μg/ml) for 24 h. *$P < 0.05$ vs. control. ($n = 6$.) Data are expressed as means ± SE. Student's *t*-test was employed for comparisons between two groups; one-way ANOVA followed by Tukey's post-test for multiple comparisons was used for groups of three or more

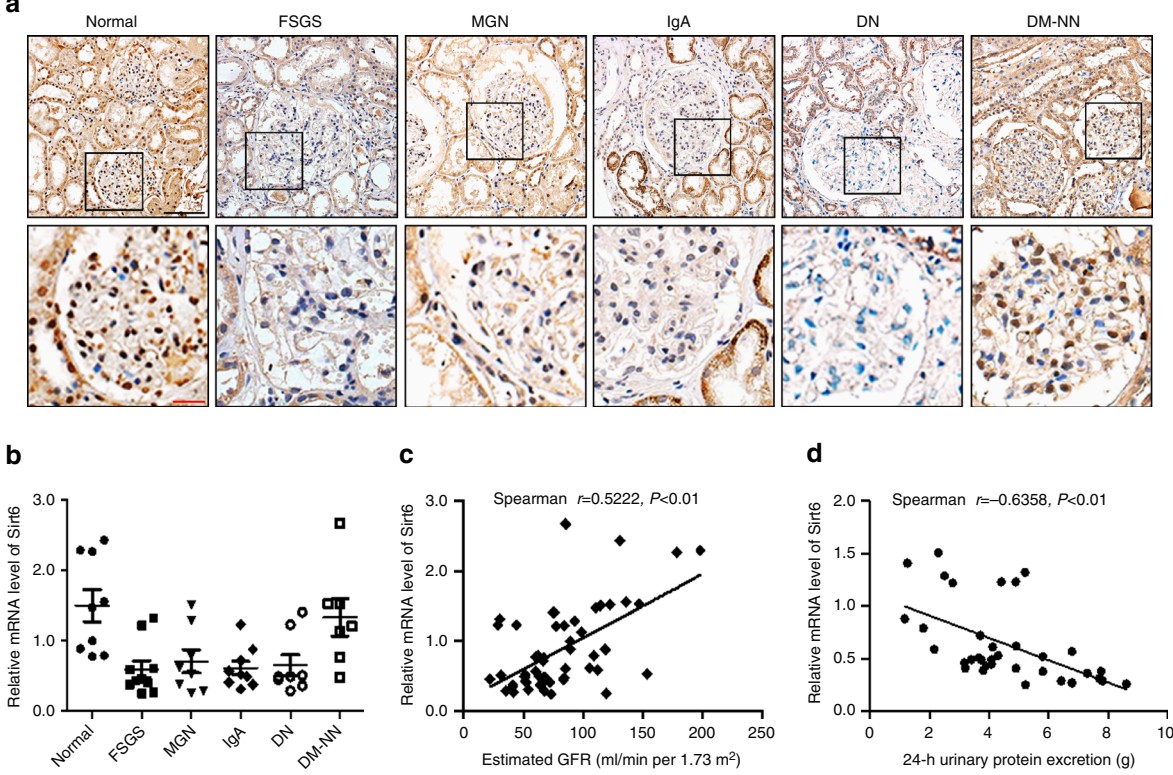

**Fig. 2** Downregulation of Sirt6 in renal biopsies from patients with different forms of podocytopathies. **a** Representative photomicrographs of Sirt6 immunohistochemical staining in human renal cortical tissue from normal subjects ($n = 9$), patients with focal segmental glomerulosclerosis (FSGS, $n = 10$), membranous glomerulonephritis (MGN, $n = 8$), IgA nephropathy (IgA, $n = 9$); diabetic nephropathy (DN) ($n = 8$), or diabetic patients without nephropathy (DM-NN). Scale bar: black 100 μm, red 25 μm. ($n = 7$). **b** Relative mRNA levels of Sirt6 in the renal biopsies from patients with different forms of podocytopathies. **c** Positive correlation (Spearman $r = 0.5222$, $P < 0.01$) between Sirt6 mRNA levels and estimated glomerular filtration rate (GFR) in all subjects. **d** Negative correlation (Spearman $r = -0.6358$, $P < 0.01$) between Sirt6 mRNA levels and proteinuria in all subjects. *$P < 0.05$ vs. normal subjects. Data are expressed as means ± SE. One-way ANOVA followed by Tukey's post-test for multiple comparisons was used for groups of three or more

(FSGS), membranous glomerulonephritis (MGN), diabetic nephropathy (DN) and IgA nephropathy. In these conditions, podocytes lose specific markers of differentiation, undergo foot process effacement and eventual detachment, and reduce the capacity to maintain the glomerular filtration barrier, thereby resulting in proteinuria[3]. Therefore, therapies aimed at preventing podocyte injury or promoting podocyte repair have major potential clinical and economic benefits. Although studies have seen dramatic advances in the understanding of podocyte biology and the pathogenesis of podocyte injury, the delivery of podocyte-specific therapies is still a great challenge. For instance, many current therapies such as glucocorticosteroids, calcineurin antagonists and mTOR inhibitors have potent effects on podocytes, but the nonspecific natures of these agents lead to undesirable systemic adverse effects[1]. Therefore, identifying the key and universal molecules involved in the different forms of podocytopathies may provide clues to develop new therapeutic strategies for patients with proteinuric kidney disease.

Histone deacetylases (HDACs) are enzymes that balance the acetylation activities of histone acetyltransferases on chromatin remodeling and play essential roles in the modulation of physiological and pathological gene transcription. Sirt6 is a member of the sirtuin family of class III NAD⁺-dependent HDACs, which consists of seven enzymes (Sirt1 to Sirt7) that share conserved core catalytic domains, but differ in their cellular localization and tissue distribution. The pleotropic identity of Sirt6 is manifested into several catalytic activities including deacetylation and ribosylation, which influence the physiology of multiple cell types and tissues. Sirt6-dependent deacetylation of

H3K9 or H3K56 is required for the regulation of genes associated with glucose/lipid metabolism, DNA repair, telomerase function, genomic stability, and cellular senescence. Sirt6-deficient mice have displayed genomic instability and several phenotypes of accelerated premature aging[4]. Sirt6 also functions as a central regulator of hematopoietic stem cell homeostasis[5] and somatic growth[6]. However, the role of Sirt6 in the kidney keeps unclear.

In this study, we found that the level of Sirt6 was reduced in renal biopsies from patients with podocytopathies and was correlated with glomerular filtration rate. Podocyte-specific deletion of Sirt6 exacerbated podocyte injury and proteinuria in two independent mouse models including DN and adriamycin (ADR)-induced nephropathy. Mechanistically, Sirt6 negatively regulated Notch signaling by the suppression of the transcription of Notch1 and Notch4 genes. Our findings suggest Sirt6 as a potential therapeutic target in proteinuric kidney disease.

## Results

**Sirt6 is reduced in podocytes from DN or ADR mice.** We first assessed the expression patterns of SIRTs in the kidney from STZ-induced diabetic mice and ADR-treated mice (a single injection of ADR leads to significant glomerular damage that recapitulates the human disease of FSGS) individually, two independent models for proteinuric kidney disease. Our results showed that the levels of Sirt1, Sirt3, Sirt4, and Sirt6 were reduced in the kidney from STZ-induced diabetic mice. In mice with ADR nephropathy, Sirt1, Sirt5, and Sirt6 were downregulated in the kidney, but the expression of Sirt4 was shown an increase

tendency, indicating different expression patterns of SIRTs under different pathological conditions (Fig. 1a). Considering that both Sirt1 and Sirt6 were reduced in DN and ADR nephropathy and recent studies have highlighted the contribution of Sirt1 to the

regulation of renal function, this study was designed to further explore the role of Sirt6 in the kidney. The reduced Sirt6 expression was also observed in the kidney from *db/db* mice (Fig. 1b), another in vivo model of type II diabetes. Furthermore,

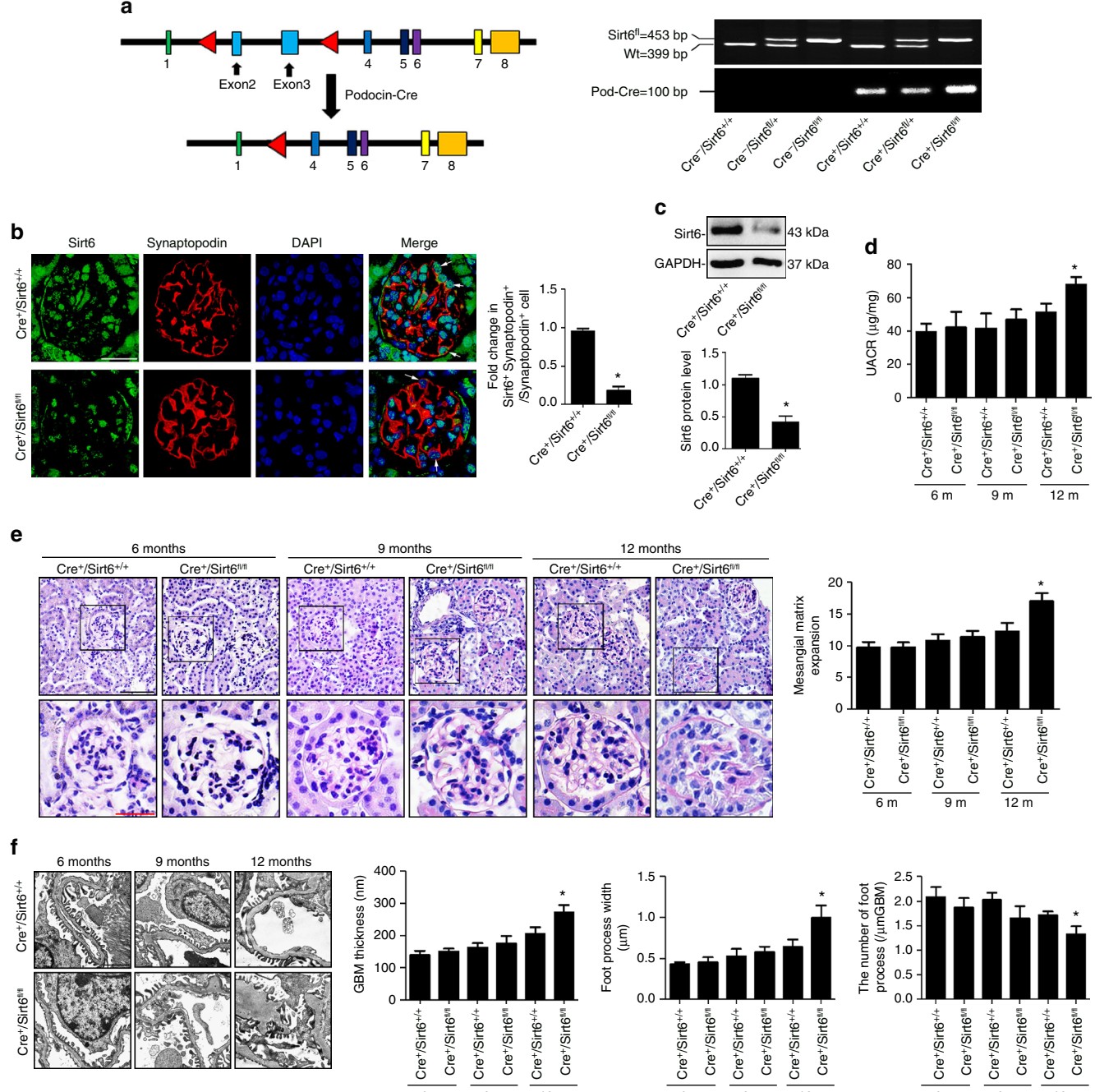

**Fig. 3** Establishment of podocyte-specific *Sirt6* knockout (*Cre+/Sirt6fl/fl*) mice. **a** Generation of conditional knockout mice in which *Sirt6* is specifically ablated in podocytes by using Cre–LoxP recombination system. Exon 2 and 3 are deleted upon *NPHS2-Cre*-mediated recombination. Genotyping was confirmed by tail preparation and PCR at 2 weeks of age. **b** Podocyte-specific loss of *Sirt6* was confirmed by immunofluorescent staining of Sirt6 in podocytes, synaptopodin was used as a podocyte marker. The *arrows* indicate representative podocytes. *Scale bar*, 25 μm. **c** Representative western blot gel documents and summarized data showing the decreased expression of Sirt6 in isolated glomeruli from podocyte-specific *Sirt6* knockout (*Cre+/Sirt6fl/fl*) mice. **d** UACR (urine albumin-to-creatinine ratio) in mice at different age. **e** Photomicrographs and quantifications showing typical glomerular structures in mice at different age. *Scale bar*: *black* 50 μm, *red* 25 μm. **f** Representative photomicrographs and quantifications of mean glomerular basement membrane (GBM) thickness, mean foot process width, and the number of foot processes in different groups of mice at different age by transmission electron microscopy (TEM) analyses. *Scale bar*, 2 μm. Podocyte-specific *Sirt6* knockout mice: *Cre+/Sirt6fl/fl* mice; Mice with two WT alleles and Cre expression (*Cre+/Sirt6+/+*) were used as controls. *$P < 0.05$ vs. control (*Cre+/Sirt6+/+* mice). ($n = 8$.) Data are expressed as means ± SE. Student's *t*-test was employed for comparisons between two groups; one-way ANOVA followed by Tukey's post-test for multiple comparisons was used for groups of three or more

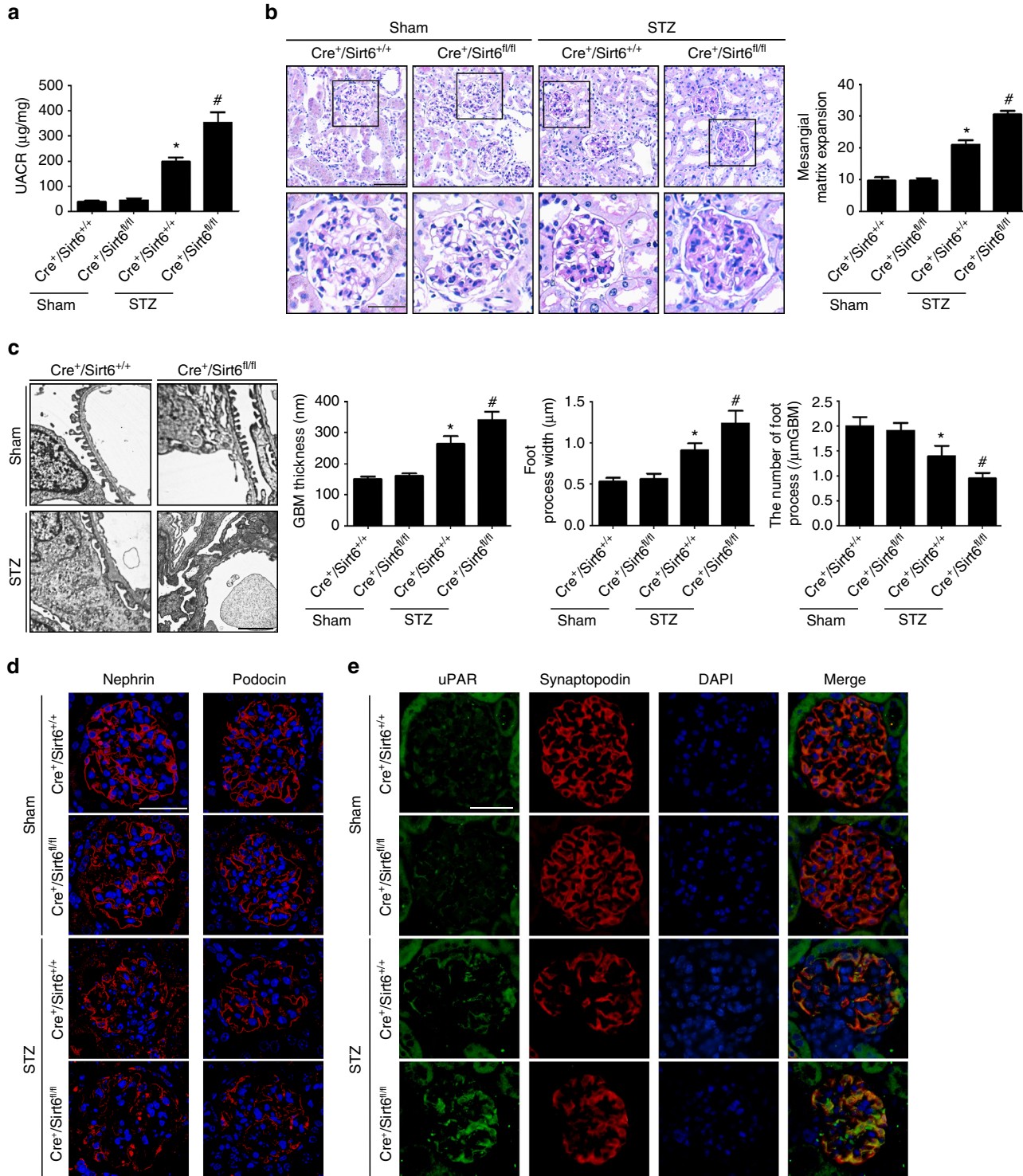

**Fig. 4** Podocyte-specific loss of *Sirt6* exacerbates podocyte injury and proteinuria in DN. **a** UACR (urine albumin-to-creatinine ratio) in different groups of mice. **b** Photomicrographs and quantifications showing typical glomerular structure changes in different groups of mice. *Scale bar*: *black* 50 μm, *red* 25 μm. **c** Representative photomicrographs and quantifications of mean glomerular basement membrane (GBM) thickness, mean foot process width, and the number of foot processes in different groups of mice at different age by transmission electron microscopy (TEM) analyses. *Scale bar*, 3 μm. **d** Representative confocal microscopic images showing the expressions of nephrin and podocin in the kidney from different groups of mice. *Scale bar*, 25 μm. **e** Representative confocal microscopic images showing the expressions of uPAR in podocytes from different groups of mice. *Scale bar*, 25 μm. Podocyte-specific *Sirt6* knockout mice: *Cre+/Sirt6fl/fl* mice; Mice with two WT alleles and Cre expression (*Cre+/Sirt6+/+*) were used as controls. *$P < 0.05$ vs. control (*Cre+/Sirt6+/+* mice), #$P < 0.05$ vs. *Cre+/Sirt6+/+* STZ mice. ($n = 8$.) Data are expressed as means ± SE. One-way ANOVA followed by Tukey's post-test for multiple comparisons was used for groups of three or more

immunofluorescent results showed a significant reduction of podocyte Sirt6 expression from STZ-induced diabetic mice and ADR-treated mice by double immunostaining of Sirt6 and podocyte specific markers synaptopodin (Fig. 1c) or Wilms' tumor protein-1 (WT-1) (Supplementary Fig. 1). In vitro, high

glucose (HG, Fig. 2d) or advanced glycation end-products (AGE, Fig. 2e) significantly reduced podocyte Sirt6 expression in a concentration-dependent manner. In addition, we also demonstrated the reduction of Sirt6 expression in ADR-treated podocytes (Fig. 1f), suggesting that the decreased Sirt6 expression may

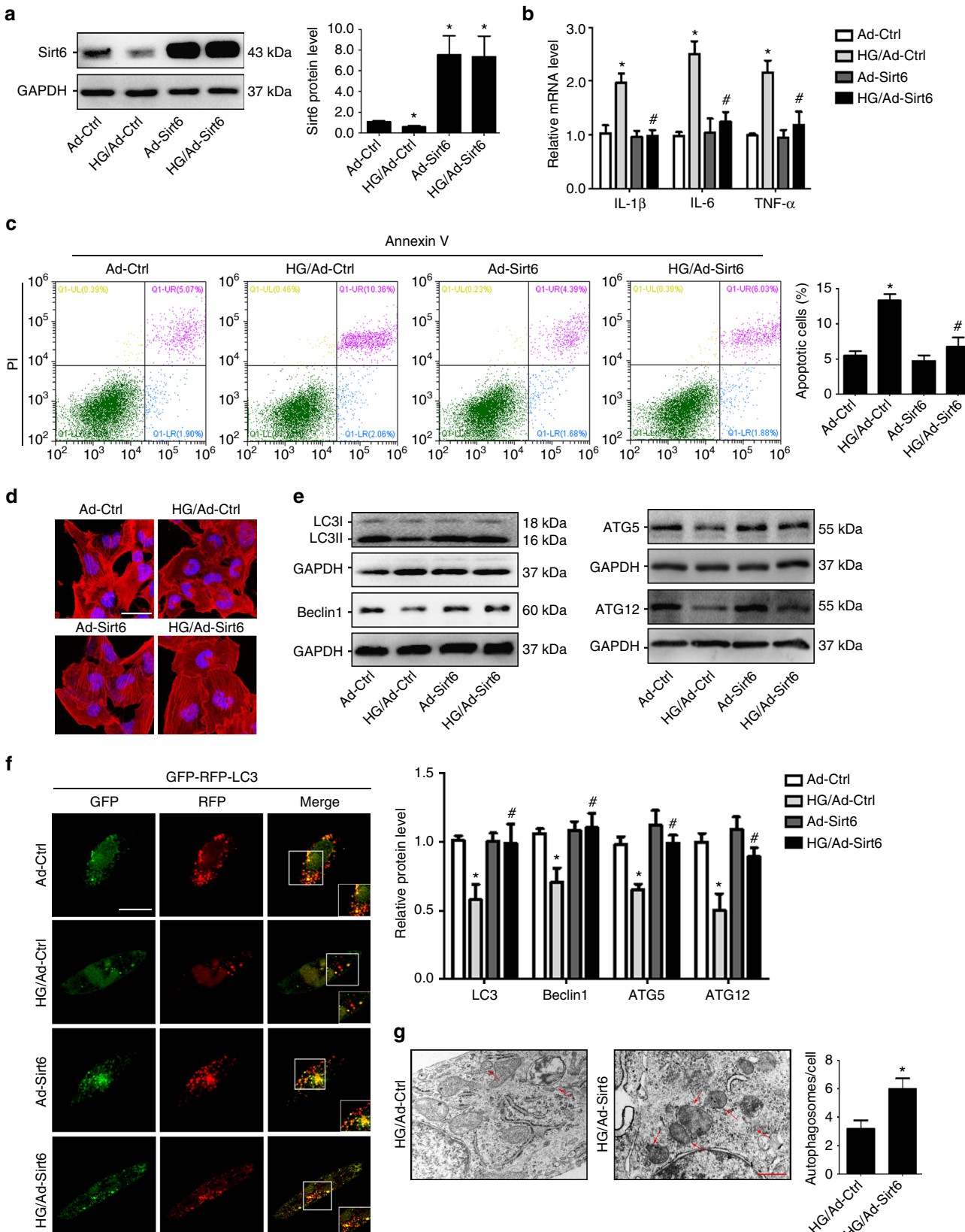

importantly contribute to podocyte injury under different pathological conditions. We also found that other renal parenchymal cells including rat glomerular mesangial cells, rat glomerular endothelial cells, and human tubule epithelial cells (HK-2) had different expression patterns of Sirt6 in response to hyperglycemia (Supplementary Fig. 2) or ADR (Supplementary Fig. 3). In addition, we also characterized the expression profile of Sirt6 in various types of renal parenchymal cells and found that Sirt6 may have different intracellular localizations in these cells (Supplementary Fig. 4).

**Reduction of Sirt6 in various podocytopathies renal biopsies.** We first confirmed the reduction of Sirt6 in renal biopsies from DN ($n = 8$) and FSGS ($n = 10$) subjects compared with normal subjects or diabetic patients without nephropathy (DM-NN, $n = 7$) by immunohistochemistry (IHC) staining (Fig. 2a) and real time reverse transcription (RT)-PCR analysis (Fig. 2b). Importantly, the Sirt6 level was also markedly reduced in renal biopsies from patients with other different forms of podocytopathies such as MGN ($n = 8$) and IgA nephropathy ($n = 9$). Notably, the mRNA levels of Sirt6 were positively correlated with estimated glomerular filtration rate (estimated GFR, Fig. 2c) and negatively correlated with proteinuria in all subjects (Fig. 2d).

**Sirt6 deletion exacerbates podocyte injury in diabetic mice.** To elucidate the role of Sirt6 in podocyte homeostasis and injury in vivo, we generated conditional knockout mice in which Sirt6 is specifically ablated in podocytes by using Cre–LoxP recombination system. Podocin-Cre mice were crossed with $Sirt6^{fl/fl}$ to generate $Podocin-Cre\ Sirt6^{fl/fl}$ mice ($Cre^+/Sirt6^{fl/fl}$ mice), which was identified by tail genotyping (Fig. 3a), immunofluorescent analysis in podocytes (Fig. 3b) and a significant reduction in the protein levels of Sirt6 in isolated glomeruli from $Cre^+/Sirt6^{fl/fl}$ mice (Fig. 3c). All mice were viable and fertile. Firstly, we analyzed $Cre^+/Sirt6^{fl/fl}$ mice and control mice in age-matched groups. Up to 9-month after birth, $Cre^+/Sirt6^{fl/fl}$ mice were indistinguishable from control littermates as analyzed by renal histology, albuminuria, and glomerular ultrastructure. In a 12-month follow-up, the ageing $Cre^+/Sirt6^{fl/fl}$ mice developed a slight albuminuria (Fig. 3d), mesangial matrix expansion (Fig. 3e) and podocyte injury as evidenced by glomerular basement membrane (GBM) thickening, and podocyte foot process broadening and effacement (Fig. 3f) but significantly higher than in control mice. Moreover, we also evaluated the expression levels of other isoforms of sirtuins in isolated glomeruli from $Cre^+/Sirt6^{fl/fl}$ mice in vivo or Sirt6-deficient podocytes in vitro. At the expression levels, we did not find other sirtuins to compensate for Sirt6 deficiency (Supplementary Fig. 5).

In Supplementary Table 1, STZ-induced diabetic mice (22-week old) had hyperglycemia and lower body weight

compared with their non-diabetic counterparts, no difference in blood pressure among these groups. As shown in Fig. 4a, diabetic $Cre^+/Sirt6^{fl/fl}$ mice exhibited a significant increase in urinary albumin excretion as compared with diabetic $Cre^+/Sirt6^{+/+}$ mice. Morphological examinations showed the glomerular mesangium expansion (Fig. 4b) and podocyte injury (Fig. 4c) from diabetic $Cre^+/Sirt6^{+/+}$ mice, all of which were exacerbated in diabetic $Cre^+/Sirt6^{fl/fl}$ mice. Podocyte injury was further confirmed by a significant loss of key podocyte differentiation markers including nephrin and podocin in diabetic $Cre^+/Sirt6^{fl/fl}$ mice (Fig. 4d). In addition, considering that the expression of urokinase plasminogen activator receptor (uPAR) is induced in podocytes during proteinuric kidney disease, which is required for the development of podocyte foot process effacement and proteinuria, we further detected the uPAR expression in different groups of mice and found the low-level of uPAR expression in podocytes from control mice by immunofluorescent analysis. In contrast, diabetic $Cre^+/Sirt6^{+/+}$ mice had a significant increase in podocyte uPAR expression, an even stronger induction of uPAR expression was found in podocytes from diabetic $Cre^+/Sirt6^{fl/fl}$ mice (Fig. 4e).

**Sirt6 has pleiotropic protective actions in podocytes.** In vitro, overexpression of Sirt6 by a Sirt6-adenovirus transfection (Fig. 5a) reduced the levels of proinflammatory mediators (Fig. 5b), attenuated podocyte apoptosis (Fig. 5c), and ameliorated actin cytoskeleton derangement (Fig. 5d) in podocytes with HG treatment. In addition, overexpression of Sirt6 also reduced the expression level of uPAR in podocytes with HG treatment (Supplementary Fig. 6). Moreover, we observed that overexpression of Sirt6 restored autophagy in podocytes as evidenced by the increased levels of autophagy-related proteins (Fig. 5e). Finally, we utilized the tandem RFP-GFP-LC3 adenovirus construct to confirm autophagy induction by form punctate that represent autophagosome formation. In Fig. 5f, we observed the successful introduction of the RFP-GFP-LC3 adenovirus construct showing both fluorescent proteins. In addition to accumulation of LC3, more red puncta were present in podocytes with Sirt6 overexpression than in controls under HG condition. In addition, the number of typical autophagosomes with double membranes was increased in podocytes with Sirt6 overexpression (Fig. 5g). These results further confirmed that Sirt6 induces autophagy in podocytes.

**Sirt6 inhibits Notch signaling by deacetylating H3K9.** Considering that Sirt6 is a highly specific deacetylase that targets H3K9 for the regulation of gene expression by modifying chromatin structure[7, 8], we then measured the levels of H3K9 acetylation (H3K9ac) under pathological conditions and found that the H3K9ac levels were significantly increased in renal biopsies from patients with DN and FSGS by IHC staining

**Fig. 5** Sirt6 reduces inflammatory responses, apoptosis and was essential for maintaining basal autophagy in podocytes with HG treatment: **a** Representative western blot gel documents and summarized data showing the relative protein level of Sirt6 by an Sirt6-adenovirus transfection. **b** The levels of pro-inflammatory mediators in podocytes with different treatments. **c** Podocytes with different treatments were stained with fluorescein isothiocyanate (FITC)-conjugated Annexin V and propidium iodide (PI), and analyzed by flow cytometry to evaluate the role of Sirt6 in the prevention of apoptosis. Quantification of the apoptotic cells was showed at *right panel*. *$P < 0.05$ vs. control, #$P < 0.05$ vs. scramble of HG treatment. ($n = 6$). **d** Representative confocal microscopic images showing that actin cytoskeleton derangement as evidenced by the loss of actin filaments and a granular cytoplasmic pattern of actin was ameliorated by overexpression of Sirt6. *Scale bar*, 40 μm. **e** Representative western blot gel documents and summarized data showing the relative protein levels of autophagy-associated proteins in podocytes with different treatments. **f** Representative images of LC3 staining by measurement of fluorescent intensity in podocytes showing LC3 staining in different groups of podocytes infected with RFP-GFP-LC3 adenovirus for 24 h. *Scale bar*, 20 μm. **g** Representative electronic micrographs and summarized data showing the number of autophagosomes/cell in different groups (a random number of 30 cells were selected for each group) including HG-treated podocytes with or without Sirt6 overexpression. The *arrows* indicate autophagosomes. *Scale bar*, 1 μm. *$P < 0.05$ vs. control, #$P < 0.05$ vs. scramble of HG treatment. ($n = 6$.) Data are expressed as means ± SE. Student's $t$-test was employed for comparisons between two groups; one-way ANOVA followed by Tukey's post-test for multiple comparisons was used for groups of three or more

(Fig. 6a). In animal studies, a more significant increase in H3K9ac in podocytes from diabetic *Cre+/Sirt6fl/fl* mice was also observed as compared with diabetic *Cre+/Sirt6+/+* mice (Fig. 6b). In vitro, HG-induced H3K9ac was attenuated by overexpression of Sirt6 (Fig. 6c). Moreover, we observed the changes of Notch signaling in HG-treated podocytes with or without Sirt6 overexpression by

Agilent Whole Human Genome Oligo Microarray for global gene expression analysis (Fig. 6d). Real time RT-PCR analysis found that among *Notch* isoforms, only *Notch1* and *Notch4* were upregulated in HG-treated podocytes, which could be attenuated by overexpression of Sirt6 (Fig. 6e). The same tendency was also found in podocytes from diabetic *Cre+/Sirt6fl/fl* mice as compared

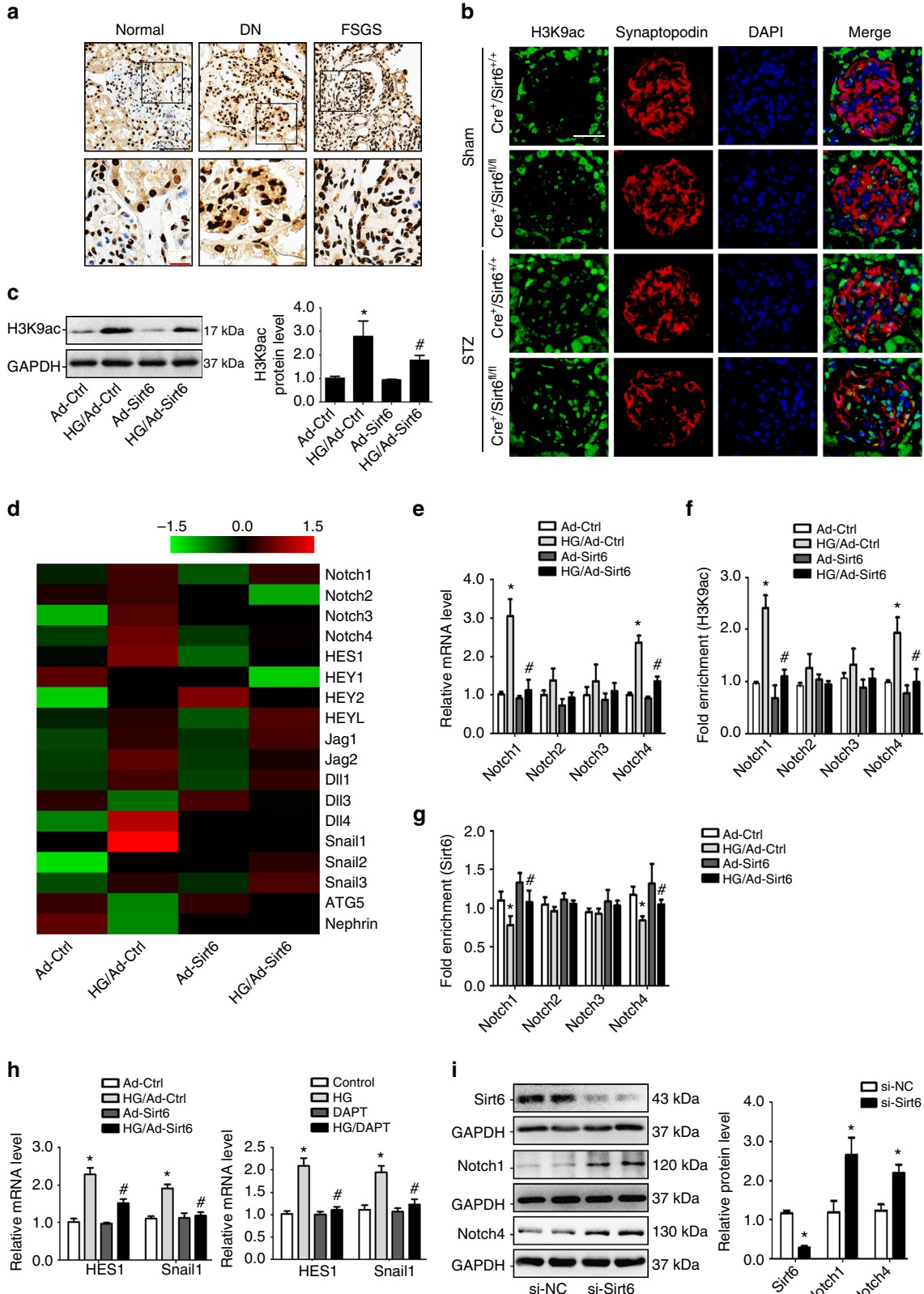

with diabetic $Cre^+/Sirt6^{+/+}$ mice (Supplementary Fig. 7). Mechanistically, we examined whether Sirt6 deficiency results in the increased H3K9ac in the promoters of Notch genes by chromatin immunoprecipitation (ChIP). Our results revealed the increased levels of H3K9ac in the promoters of Notch1 and Notch4 in podocytes with HG treatment (Fig. 6f). To confirm whether Sirt6 can recruit and deacetylate H3K9 on these promoters, we performed ChIP assay in podocytes with over-expression of Sirt6 and found that Sirt6 significantly reduced the level of H3K9ac in the promoter region of Notch1 and Notch4 (Fig. 6f). Our results further confirmed that Sirt6 could bind to the promoter of Notch1 and Notch4 (Fig. 6g). Under HG condition, Sirt6 was dissociated from the promoters of these genes in podocytes and recruited to them after Sirt6-adenovirus transfection by ChIP analysis (Fig. 6g). Furthermore, over-expression of Sirt6 reduced HG-induced Notch signaling downstream target genes such as HES1 and Snail1 by mRNA analysis as well as the results from podocytes treated with DAPT that is a γ-secretase inhibitor and indirectly inhibit Notch signaling pathways (Fig. 6h), which was consistent with in vivo studies showing more significant increases in the levels of HES1 and Snail1 in podocytes from diabetic $Cre^+/Sirt6^{fl/fl}$ mice (Supplementary Fig. 7). Moreover, we found that gene silencing of Sirt6 can regulate the basal levels of Notch1 and Notch4 in podocytes (Fig. 6i).

**Notch inhibition rescues Sirt6-deficient podocytes injury.** Inhibition of Notch signaling reduced inflammatory responses (Fig. 7a) and apoptosis (Fig. 7b), ameliorated actin cytoskeleton derangement (Fig. 7c), and recovered the expression levels of nephrin and podocin (Fig. 7d) in Sirt6-deficient podocytes. As well as gene silencing of Notch1 restored the levels of autophagy-associated proteins in podocytes with HG treatment (Supplementary Fig. 8), gene silencing of Notch1 can also restore the levels of autophagy-associated proteins (Fig. 7e) and autophagy flux (Fig. 7f) in Sirt6-deficient podocytes.

**Sirt6 deletion exacerbates podocyte injury in ADR mice.** We also utilized the murine model of ADR nephropathy to confirm the role of Sirt6 in maintaining podocyte function. ADR-treated $Cre^+/Sirt6^{fl/fl}$ mice exhibited significant increases in urinary albumin excretion (Fig. 8a), glomerulosclerosis (Fig. 8b) and podocyte injury (Fig. 8c), and marked decreases in the levels of nephrin and podocin (Fig. 8d) compared with the ADR-treated $Cre^+/Sirt6^{+/+}$ mice. Furthermore, we found that overexpression of Sirt6 attenuated ADR-induced podocyte apoptosis (Fig. 8e). Sirt6 was also confirmed to regulate H3K9ac at promoters of Notch1 and Notch4 in podocytes with ADR treatment (Fig. 8f, g). Through regulating H3K9ac at promoters of Notch1 and Notch4, Sirt6 reduced Notch1 and Notch4 transcription (Fig. 8h). Significant increases in H3K9ac, Notch1 and Notch4 expression

were also observed in podocytes from ADR $Cre^+/Sirt6^{fl/fl}$ mice (Supplementary Fig. 9).

**Overexpression of Sirt6 ameliorates renal injury in vivo.** To examine genetic therapeutic efficiency targeting to Sirt6 in ADR nephropathy, recombinant lentivirus vectors pGLV3 harboring Sirt6 were delivered into the kidney of $Cre^+/Sirt6^{fl/fl}$ mice by means of intraparenchymal injections. Our results confirmed the efficiency of in vivo gene transfer showing that mice received Sirt6 lentivirus markedly increased the level of Sirt6 in isolated glomeruli by mRNA analysis (Fig. 9a). Overexpression of Sirt6-significantly ameliorated renal injury as evidenced by reduced albuminuria (Fig. 9b), decreased mesangial expansion (Fig. 9c) and ameliorated podocyte injury (Fig. 9d) in $Cre^+/Sirt6^{fl/fl}$ mice with ADR treatment, as well as recovered nephrin and podocin expression (Fig. 9e) and increased the levels of LC3 and beclin1 by IHC staining (Supplementary Fig. 10). In addition, ChIP assay in isolated glomeruli further confirmed that overexpression of Sirt6 deacetylated H3K9ac at promoter of Notch1 (Fig. 9f, g). Consistently, the levels of Notch1, Notch4 and H3K9ac were significantly decreased in podocytes after transfection with Sirt6 lentivirus in $Cre^+/Sirt6^{fl/fl}$ ADR-treated mice by immuno-fluorescent analysis (Supplementary Fig. 11).

## Discussion

Although emerging evidence have indicated the importance of epigenetic modification on the regulation of renal function[9, 10], current understanding of biological functions of sirtuins in the kidney is very limited and their roles in epigenetic-mediated renal functions remain to be elucidated. In this study, we found that both Sirt1 and Sirt6 were downregulated in the kidney from mice with DN or ADR nephropathy, two independent models for proteinuric kidney disease. Considering that recent studies highlight the contribution of Sirt1 to the regulation of renal function[11-14], our studies was designed to further explore the functional role of Sirt6 in the kidney. Firstly, we characterized the expression profile of Sirt6 in various types of renal parenchymal cells such as rat glomerular mesangial cells, rat glomerular endothelial cells, human podocytes, and human proximal tubule epithelial cells (HK-2). It was found that Sirt6 was expressed in all these major types of renal cells. Although Sirt6 was originally identified as a nuclear-localizing protein[15] and our studies showed that Sirt6 localized in the nucleus of human podocytes, Sirt6 was also found more or less in the cytoplasm of some other renal cells by our current antibody-based molecular assays, leading us to speculate that renal Sirt6 may shuttle between the nucleus and cytoplasm, depending on cell type or environmental stimuli. One of the most important findings in this study was a significant reduction in Sirt6 expression in the kidney that was correlated with estimated GFR and the levels of proteinuria from

**Fig. 6** Sirt6 modulates Notch signaling by deacetylating H3K9ac at promoters of Notch1 and Notch4. **a** Representative photomicrographs of H3K9ac immunohistochemical staining in human renal tissue from normal subjects and patients with focal segmental glomerulosclerosis (FSGS) and diabetic nephropathy (DN). Scale bar: black 100 μm, red 25 μm. **b** Representative confocal microscopic images showing the levels of H3K9ac in podocytes of the kidney from different groups of mice, synaptopodin was used as a podocyte marker. Scale bar, 25 μm. **c** Representative western blot gel documents and summarized data showing the effect of overexpression Sirt6 on the levels of H3K9ac in podocytes treated with high glucose (HG, final concentration 40 mmol/l in medium) for 24 h. **d** Representative heatmap of gene expression levels by multiplex quantitative RT-PCR array. **e** Summarized data showing the effect of Sirt6 on the mRNA levels of Notch genes in podocytes treated with HG. **f** Chromatin immunoprecipitation (ChIP) analysis showing the acetylation levels of H3K9 (H3K9ac) in the promoters of Notch1 and Notch4 using antibodies to H3K9ac. **g** ChIP analysis showed that Sirt6 bound to promoters of Notch1 and Notch4 genes by using antibodies to Sirt6 in podocytes. Sirt6 was dissociated from the promoters of these genes in podocytes with HG treatment and recruited to them after overexpression of Sirt6. *$P < 0.05$ vs. control, #$P < 0.05$ vs. scramble of HG treatment. ($n = 6$.) **h** Summarized data showing the effect of overexpression of Sirt6 or inhibition of Notch signaling by DAPT on the mRNA levels of Notch downstream genes HES1 and Snail1 in podocytes treated with HG. **i** Representative western blot gel documents and summarized data showing Sirt6 deficiency on the effect of Notch1 and Notch4 expressions in podocytes. *$P < 0.05$ vs. control. ($n = 6$.) Data are expressed as means ± SE. Student's t-test was employed for comparisons between two groups; one-way ANOVA followed by Tukey's post-test for multiple comparisons was used for groups of three or more

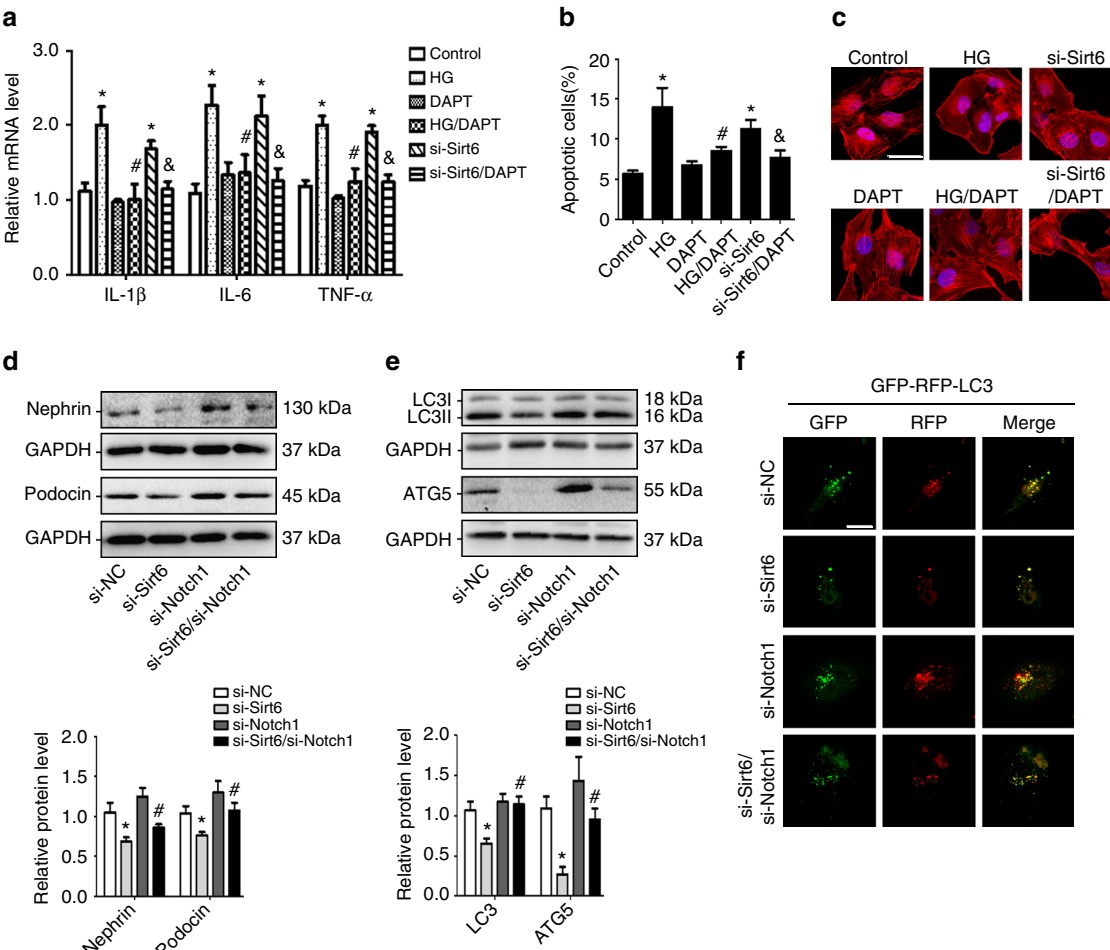

**Fig. 7** Inhibition of the Notch pathway rescues the functional defect in Sirt6-deficient podocytes. **a** The relative levels of pro-inflammatory mediators in podocytes with different treatments. **b** Summarized data showing the level of podocyte apoptosis determined by flow cytometric analysis in podocytes with different treatments. Podocytes were stained with fluorescein isothiocyanate (FITC)-conjugated Annexin V and propidium iodide (PI), and analyzed by flow cytometry. *$P < 0.05$ vs. control, #$P < 0.05$ vs. scramble of HG treatment, &$P < 0.05$ vs. siRNA-transfected podocytes. ($n = 6$.) **c** Representative confocal microscopic images showing that actin cytoskeleton derangement as evidenced by the loss of actin filaments and a granular cytoplasmic pattern of actin was ameliorated by DAPT with different treatments. *Scale bar*, 40 μm. **d** Representative western blot gel documents and summarized data showing that gene silencing of *Notch1* restored the expressions of nephrin and podocin in Sirt6-deficient podocytes ($n = 6$). **e** Representative western blot gel documents and summarized data showing that gene silencing of *Notch1* rescued the autophagy defect in Sirt6-deficient podocytes ($n = 6$). **f** Representative images of LC3 staining by measurement of fluorescent intensity in podocytes showing LC3 staining in different groups of podocytes infected with RFP-GFP-LC3 adenovirus for 24 h. *Scale bar*, 20 μm. *$P < 0.05$ vs. si-negative control, #$P < 0.05$ vs. si-Sirt6 transfected podocytes. ($n = 6$.) Data are expressed as means ± SE. One-way ANOVA followed by Tukey's post-test for multiple comparisons was used for groups of three or more

patients with different types of podocytopathies, suggesting that the reduction of Sirt6 may be a common feature of human proteinuric kidney disease. To explore the role of Sirt6 on the regulation of podocyte function, we established podocyte-specific Sirt6 knockout ($Cre^+/Sirt6^{fl/fl}$) mice and renal injury was induced by STZ and ADR individually. The increases in urinary albumin excretion and the severity of glomerular injury were all significantly greater in $Cre^+/Sirt6^{fl/fl}$ mice than in control mice, indicating that Sirt6 as a central target molecule protects against podocyte injury.

Functionally, except that Sirt6 had pleiotropic protective actions such as anti-inflammation and anti-apoptosis, we also found that Sirt6 can modulate uPAR signaling in podocytes. Studies have demonstrated uPAR can lead to foot process effacement and urinary protein loss in puromycin aminonucleoside nephrosis and LPS-induced transient proteinuria in mice, which was also identified in a majority of patients with FSGS as a soluble factor that acts through binding to and activating podocyte β3

integrin to promote cell motility and kidney permeability[16, 17]. In addition, studies have showed that nuclear factor of activated T cells (NFAT) 2 inhibitor ameliorates DN and podocyte injury in db/db mice by inhibiting the expression of uPAR[18]. In this study, we found that $Cre^+/Sirt6^{fl/fl}$ diabetic mice exhibited a marked increase in the level of uPAR as compared with the $Cre^+/Sirt6^{+/+}$ diabetic mice, indicating that Sirt6 may be one of the critical components that links uPAR to podocyte injury in the pathogenesis of DN.

Emerging evidence has indicated that impaired autophagic activity is involved in the development of chronic kidney disease[19]. Podocytes display a high level of basal autophagy, which is important for the maintenance of podocyte homeostasis[20]. Podocyte specific deletion of ATG5 leads to proteinuria and glomerulopathy in aging mice[19, 21]. Although mounting evidence indicates that HDAC-mediated acetylation can regulate autophagy, different isoforms have different effects on autophagy. HDAC10 promotes autophagy-mediated cell survival[22], whereas

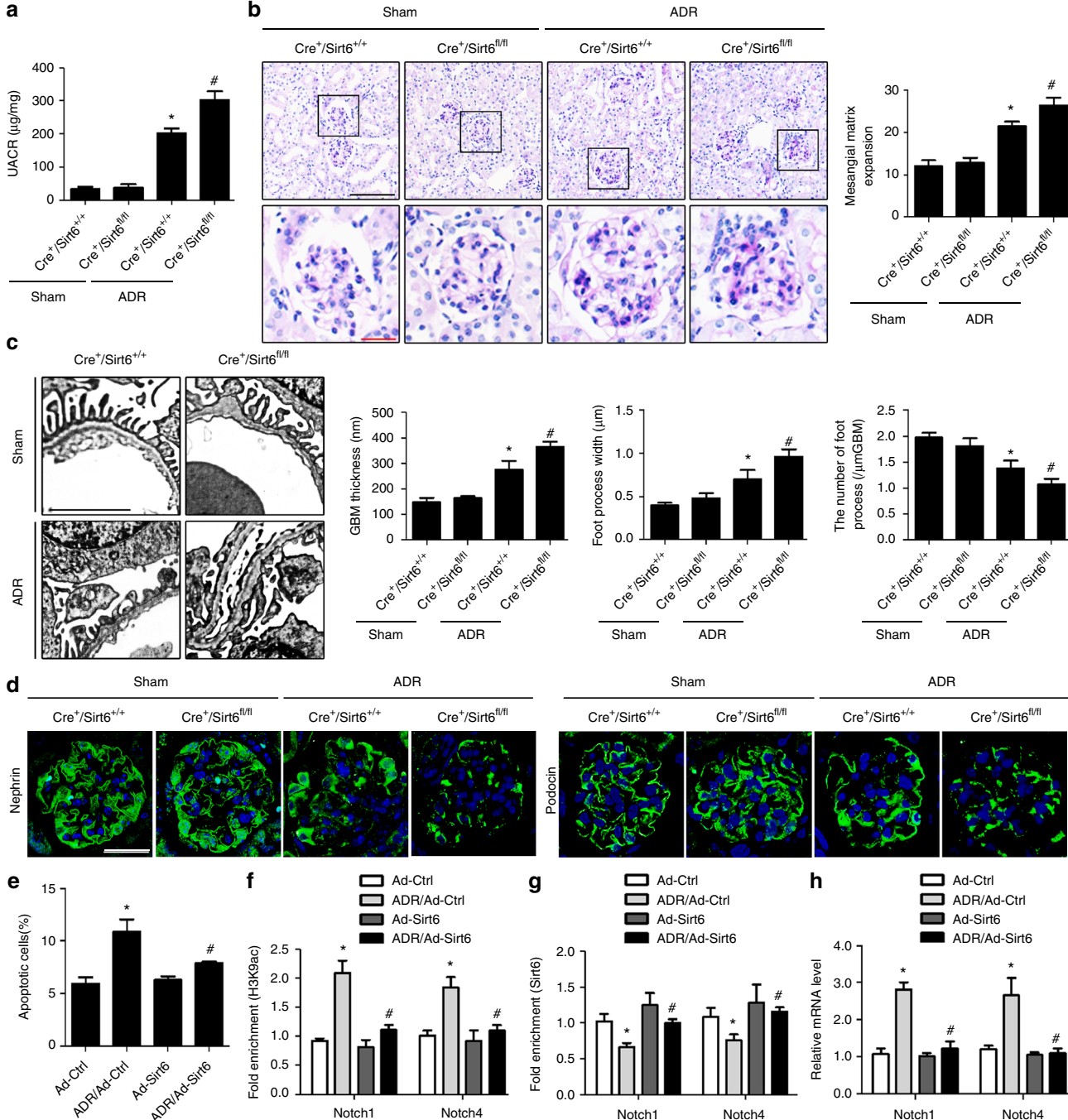

**Fig. 8** ADR-treated Podocin-*Cre Sirt6*<sup>fl/fl</sup> mice exhibites substantial podocyte injury and glomerulosclerosis. **a** UACR (urinary albumin-to-creatinine ratio) in different groups of mice. **b** Photomicrographs and quantifications showing typical changes in glomerular structure in different groups of mice. *Scale bar*: *black* 100 μm, *red* 25 μm. **c** Representative photomicrographs and quantifications of mean glomerular basement membrane (GBM) thickness, mean foot process width, and the number of foot processes in different groups of mice. *Scale bar*, 3 μm. **d** Representative confocal microscopic images showing the expressions of nephrin and podocin in podocytes of the kidney from different groups of mice. *Scale bar*, 25 μm. Podocyte-specific *Sirt6* knockout mice: *Cre*<sup>+</sup>/*Sirt6*<sup>fl/fl</sup> mice; Mice with two WT alleles and Cre expression (*Cre*<sup>+</sup>/*Sirt6*<sup>+/+</sup>) were used as controls. *$P < 0.05$ vs. control (*Cre*<sup>+</sup>/*Sirt6*<sup>+/+</sup> mice), #$P < 0.05$ vs. *Cre*<sup>+</sup>/*Sirt6*<sup>+/+</sup> ADR mice. ($n = 8$.) **e** Podocytes with different treatments were stained with fluorescein isothiocyanate (FITC)-conjugated Annexin V and propidium iodide (PI), and analyzed by flow cytometry. Summarized data showing podocyte apoptosis determined by flow cytometric analysis in podocytes with different treatments. **f** ChIP analysis showing the acetylation levels of H3K9 (H3K9ac) in the promoters of *Notch1* and *Notch4* using antibodies to H3K9ac. **g** ChIP analysis showed that Sirt6 bound to promoters of *Notch1* and *Notch4* genes by using antibodies to Sirt6 in podocytes. **h** Summarized data showing the effect of Sirt6 on the mRNA levels of *Notch1* and *Notch4* genes in podocytes treated with ADR. *$P < 0.05$ vs. control, #$P < 0.05$ vs. scramble of ADR treatment. ($n = 6$.) Data are expressed as means ± SE. One-way ANOVA followed by Tukey's post-test for multiple comparisons was used for groups of three or more

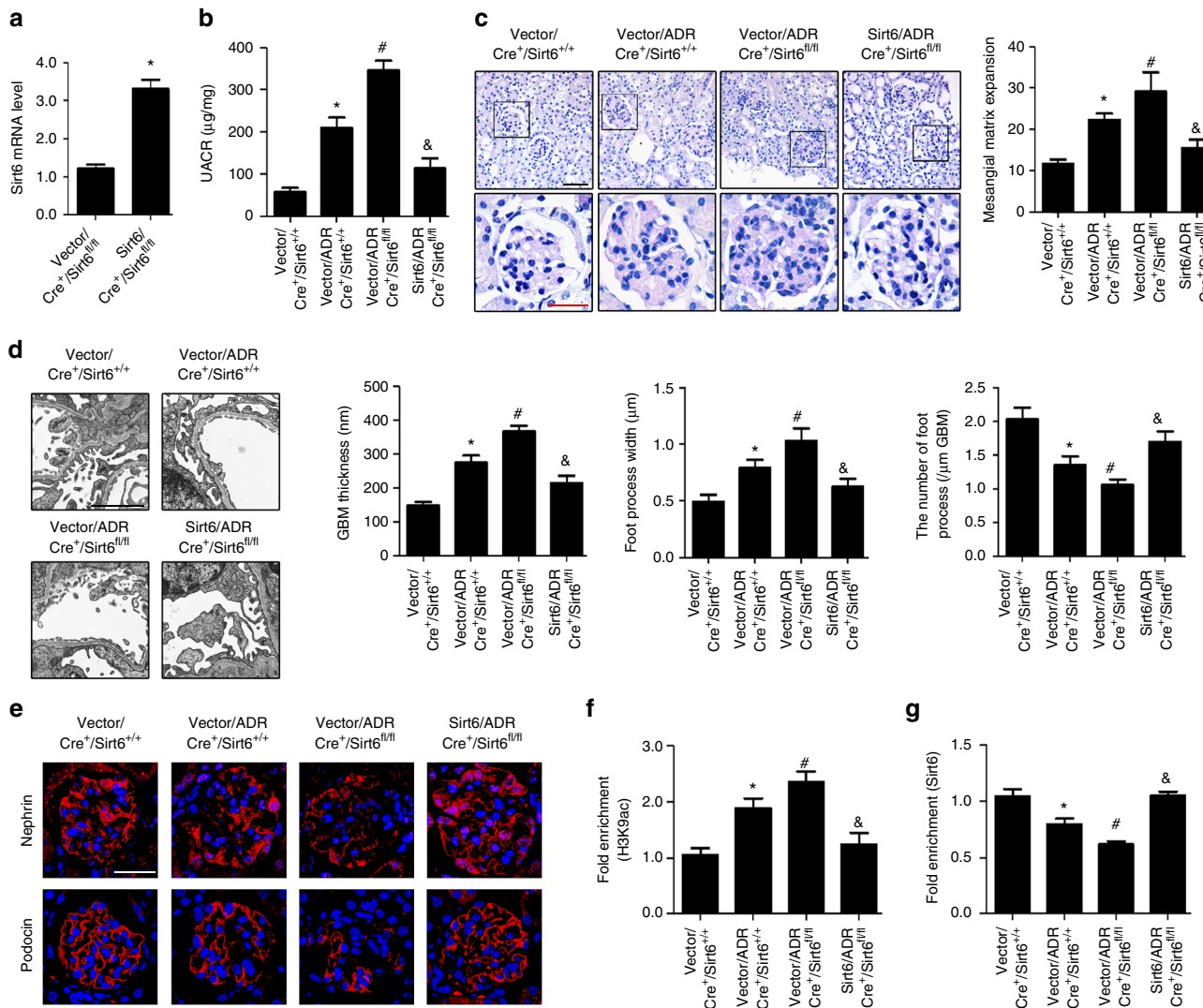

**Fig. 9** In vivo overexpression of Sirt6 by intrarenal lentiviral gene delivery ameliorates renal injury in $Cre^+/Sirt6^{fl/fl}$ mice with ADR nephropathy. **a** The relative mRNA levels of Sirt6 in isolated glomeruli at 5-week after pGLV3-Sirt6 delivered into the mouse kidney by means of intraparenchymal injections. **b** UACR (urine albumin-to creatinine ratio) in different groups of mice. **c** Photomicrographs and quantifications showing typical changes in glomerular structure in different groups of mice. *Scale bar: black* 50 μm, *red* 25 μm. **d** Representative photomicrographs and quantifications of mean glomerular basement membrane (GBM) thickness, mean foot process width, and the number of foot processes in different groups of mice. *Scale bar,* 2 μm. **e** Representative confocal microscopic images showing the expressions of nephrin and podocin in podocytes from different groups of mice. *Scale bar,* 25 μm. **f** ChIP analysis showing the acetylation levels of H3K9 (H3K9ac) in the promoter of *Notch1* using antibodies to H3K9ac in isolated glomeruli. **g** ChIP analysis showed that Sirt6 bound to promoters of *Notch1* by using antibodies to Sirt6 in isolated glomeruli. *$P < 0.05$ vs. control (vector/$Cre^+/Sirt6^{+/+}$ mice), #$P < 0.05$ vs. vector/ADR/$Cre^+/Sirt6^{+/+}$ mice, &$P < 0.05$ vs. vector/ADR/$Cre^+/Sirt6^{fl/fl}$ mice. ($n = 8$.) Data are expressed as means ± SE. Student's *t*-test was employed for comparisons between two groups; one-way ANOVA followed by Tukey's post-test for multiple comparisons was used for groups of three or more

HDAC4 negatively regulates podocyte autophagy[9]. Sirt1 induces autophagy during caloric restriction, which is associated with longevity[23]. In contrast, negative regulation of autophagy via Sirt1 was observed in the setting of cigarette smoke exposure[24]. The mixed observation may result from different substrates of individual HDACs and different contributions in different pathophysiological conditions. In this study, we attempted to clarify whether Sirt6 is involved in the regulation of autophagy in podocytes. In consistent with previous studies showing the induction of autophagy via Sirt6 in human bronchial epithelial cells[25], we found that Sirt6 positively regulated autophagy and demonstrated under pathogenic conditions, relative insufficiency of autophagy was a critical determination of podocyte injury, which was attributed to the reduced Sirt6 expression.

Considering that the changes of Notch signaling were observed in HG-treated podocytes with or without Sirt6 overexpression, and Notch signaling mediates inflammatory responses[26], cellular apoptosis[27] and autophagy[28], which can be regulated by epigenetic modification[29], Notch may be a potential target of Sirt6. The Notch pathway consists of four Notch receptors (Notch1–4) and five Notch ligands including Jagged1, Jagged2, Delta-like (Dll) 1, Dll3, and Dll4. In the kidney, Notch is a critical regulator of kidney development and this pathway is mostly silenced once kidney maturation. However, Notch signaling can be reactivated under different renal pathological conditions[30]. The Notch pathway in mature podocytes plays a critical role in the development of glomerular disease[31] and can be activated by HG[32] or other stimuli[33]. Studies have observed that conditional

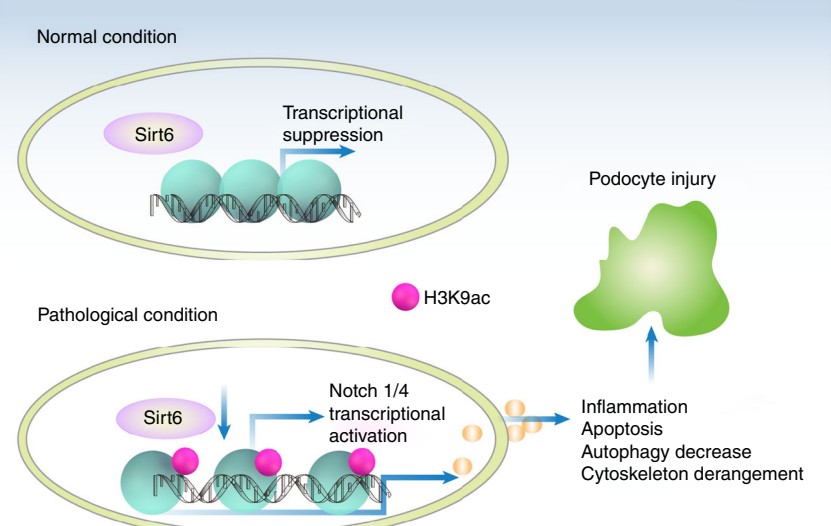

**Fig. 10** Schematic depicting *Sirt6* deficiency exacerbates podocyte injuries and proteinuria through targeting Notch signaling. Under normal conditions, SIRT6 inhibits the transcription of *Notch1* and *Notch4* genes by decreasing H3K9ac levels in the promoter region of *Notch1* and *Notch4*. Under pathological conditions, SIRT6 reduction leads to the increased levels of H3K9ac in the promoters of *Notch1* and *Notch4*, thereby enhancing the transcription of *Notch1* and *Notch4* gene. The activation of Notch signaling finally results in podocyte injury through inducing inflammation, apoptosis, actin cytoskeleton derangement, as well as the inhibition of autophagy

re-expression of the intracellular domain of Notch (NICD) exclusively in podocytes caused proteinuria and glomerulosclerosis. At the molecular levels, Notch signaling can induce endocytosis of nephrin, thereby triggering the onset of proteinuria[34]. Although precise mechanisms of epigenetic regulation at Notch target genes are not completely understood, emerging evidence suggests that histone modifications play a pivotal role in this process[35, 36]. In this study, we discovered a novel role of Sirt6 in Notch signaling. By binding to their promoters and deacetylating H3K9 at these sites, Sirt6 reduced the expression of Notch1 and Notch4, which was in consistent with the results showing that the levels of Notch1 and Notch4 were significantly increased in podocytes from *Cre+/Sirt6fl/fl* mice with DN or ADR nephropathy. Of many Notch downstream target genes, *HES1* and *Snail* are well characterized and mostly relevant to proteinuria in nephropathy[37]. In this study, we further found that podocyte Snail and HES1 expressions were markedly induced in response to HG and ADR, and Sirt6 substantially suppressed their induction. In addition, inhibition of the Notch pathway by pharmacological or genetic manipulation rescued the functional defect in *Sirt6*-deficient podocytes. We further provided direct evidence showing that overexpression of Sirt6 had therapeutic potential for ADR nephropathy through targeting Notch signaling. In this study, we cannot exclude other downstream molecules of Sirt6 may also contribute to the regulation of podocyte function. For instance, Wnt/β-catenin signaling is an evolutionarily conserved developmental signaling cascade that exhibits a pivotal function in promoting podocyte dysfunction and albuminuria in proteinuric kidney diseases[38]. Studies have demonstrated that SIRT6 controls hematopoietic stem cell homeostasis through epigenetic regulation of Wnt signaling[5], it is possible that Sirt6-mediated Wnt signaling is also associated with the regulation of podocyte function. Therefore, it is necessary to further clarify the role of Wnt signaling as well as other potential downstream targets epigenetically regulated by Sirt6 in podocytes.

It should be noted that although we did not attempt to explore molecular mechanisms of Sirt6 regulation in podocytes under pathological conditions, recent studies have reported that Sirt6 can be regulated by post-transcriptionally at the RNA and protein levels[39]. MiR-33 and miR-122 reduce Sirt6 mRNA stability[40, 41]. The MDM2 ubiquitylate can target Sirt6 protein for protease-dependent degradation[42]. *Sirt6* gene expression is also modulated by several transcription factors such as E2 transcription factor 1 (E2F1) and Runt-related transcription factor 2 (RUNX2)[43, 44]. In addition, Sirt1 can regulate Sirt6 by forming a complex with FOXO3a and NRF1 on the promoter of *Sirt6*[45]. In fact, our preliminary studies have also found that the downregulation of Foxo3a and NRF1 in podocytes with HG treatment. Further studies are needed to clarify the precise mechanisms of Sirt6 regulation.

Collectively, our studies for the first time demonstrate that Sirt6 protects against podocyte injury and proteinuria, at least in part through epigenetic regulation of Notch signaling (Fig. 10). Pharmacological targeting of Sirt6-mediated Notch signaling pathways at multiple levels may provide a novel approach for the treatment of proteinuric kidney disease.

## Methods

**Human renal biopsy samples**. Renal biopsies had been performed as part of routine clinical diagnostic investigation and collected as described in Supplementary Table 2. The samples of renal biopsies were obtained from Department of Pathology, Shandong University School of Medicine, and Department of Nephrology, Union Hospital, Tongji Medical College, Huazhong University of Science and Technology. Control samples were obtained from the healthy kidney poles of individuals who underwent tumor nephrectomies without diabetes or renal disease. Among them, diabetic patients without nephropathy (DM-NN) were selected from patients who underwent nephrectomy for solitary renal cell carcinoma and had a concomitant diagnosis of type 2 diabetes. Histologic examinations and biochemical analysis (urine albumin-to-creatinine ratio <30 mg/g) revealed no features of DN or other renal disease except for the solitary renal cell carcinoma. The investigations were conducted in accordance with the principles of the Declaration of Helsinki and were approved by the Research Ethics Committee of Shandong University after informed consent was obtained from the patients.

**Generation of podocyte-specific *Sirt6* knockout mice**. Floxed *Sirt6* mice with FVB/N background (*Sirt6fl/fl*, Jackson Laboratory, Bar Harbor, ME) were back-crossed with C57BL/6 mice more than 12 generations to produce congenic strains. Then C57BL/6 *Sirt6fl/fl* mice were crossed with mice expressing Cre recombinase (Cre) under the control of the podocin promoter (B6.Cg-Tg [*NPHS2-cre*] 295Lbh/J; Jackson Laboratory) to generate podocyte-specific *Sirt6* knockout mice (*Podocin-Cre Sirt6fl/fl* mice; *Cre+/Sirt6fl/fl* mice). Mice with two WT alleles and Cre

expression were used as controls (*Podocin-Cre Sirt6*$^{+/+}$; *Cre*$^{+}$/*Sirt6*$^{+/+}$). Genotyping by tail preparation and PCR were performed at 2 weeks of age.

**STZ-induced diabetic nephropathy (DN) in mice**. Mice with a C57BL/6 background do not develop lesions of DN readily after the induction of diabetes by STZ. In addition, a high-fat diets (HFD) provides a commonly used approach to induce obesity and insulin resistance in C57BL/6 strain. In particular, after the onset of metabolic syndrome, mice on a HFD develop increased albuminuria and glomerular lesions[46, 47]. Therefore, we used a STZ-treated uninephrectomized mice model on a HFD to hasten the development of DN following previous studies[47]. To induce DN, *Cre*$^{+}$/*Sirt6*$^{+/+}$, and *Cre*$^{+}$/*Sirt6*$^{fl/fl}$ mice with a C57BL/6 background mice (male, 10 weeks of age) were injected STZ (80 mg per kg body weight intraperitoneally daily for 3 days). All male mice had unrestricted access to food/water and were maintained for 12 weeks of HFD feeding in accordance with the institutional animal care and use committee procedures of Shandong University. All protocols were approved by the Institutional Animal Care and Use Committee of School of Medicine, Shandong University (Document No. LL-201501025) and conducted in accordance with the National Institutes of Health Guide for the Care and Use of Laboratory Animals. Different groups were allocated in a randomized manner and investigators were blinded to the allocation of different groups when doing surgeries and doing outcome evaluations.

**Adriamycin (ADR)-induced nephropathy in mice**. In the ADR model, male mice (10 weeks of age) were administered ADR (20 mg/kg) intravenously by tail vein injection[48]. Urine was collected weekly to assess for albuminuria, and mice were sacrificed 5 weeks after treatment. Different groups were allocated in a randomized manner and investigators were blinded to the allocation of different groups when doing surgeries and doing outcome evaluations.

***db/db* mice**. Twelve-week-old male type 2 diabetic *db/db* mice and genetic control *db/+* mice were obtained from the Jackson laboratory.

**Intrarenal lentivirus delivery**. After 1 day ADR injection, recombinant lentivirus vectors pGLV3-harboring *Sirt6* (pGLV3-*Sirt6*) or their negative controls (pGLV3-null) were delivered into the mouse kidney. The procedure for lentivirus delivery was performed as described with slight modifications[49]. In anesthetized mice, after temporary occlusion of left renal pedicle, a 31G needle was inserted at the lower pole of the kidney parallel to the long axis and was carefully pushed toward the upper pole. As the needle was slowly removed, 100 μl filter-purified lentivirus cocktail ($1 \times 10^5$ IU/μ1) or saline was injected. Preliminary studies showed that lentiviral-mediated mRNA expression in kidney parenchyma were significantly upregulated after 48 h and no toxicity was observed in rats treated with the lentiviral vector.

**Transmission electron microscopy (TEM)**. Electron microscopic sample handling and detection were performed by the electron microscopic core lab of Shandong University as described[47]. TEM images were analyzed using Image J (National Institutes of Health, NIH, Bethesda, MD, USA). The GBM thickness, foot process width and the number of foot processes per μm of GBM were calculated using a curvimeter (SAKURAI CO., LTD, Tokyo, Japan). Five glomeruli were randomly selected from each mouse and ten electron micrographs were taken in each glomerulus.

**Cell culture and treatments**. Rat glomerular mesangial cells were obtained from the American Type Culture Collection (ATCC, Manassas, VA, USA), human podocytes were originally provided by Dr Saleem MA from University of Bristol at UK, rat glomerular endothelial cells were obtained from Creative Bioarray (Shirley, NY, USA), and human tubule epithelial cells (HK-2) were obtained from ATCC. Rat glomerular mesangial cells were cultured in Dulbecco's modified Eagle's medium containing 18 mmol/l sodium bicarbonate, 25 mmol/l glucose. Human podocytes were cultured in RPMI 1640 medium containing 11.0 mmol/l glucose as described. Rat glomerular endothelial cells were maintained in RPMI 1640 containing 200 mg/l glucose. Human tubule epithelial cells (HK-2) were cultured in a 50:50 mixture of Dulbecco's Modified Eagle's Medium/F12. Overexpression of Sirt6 by a *Sirt6*-adenovirus transfection was used in this study. Different stimuli were used in this study: (1) HG (a final concentration of 20 or 40 mmol/l in culture medium); (2) Advanced glycation end product (AGE, 50–200 μg/ml); (3) ADR (0.4 μg/ml)

**Chromatin immunoprecipitation (ChIP) assay**. ChIP was performed according to the manufacturers' instructions by using Magna ChIP HiSens Kit (Millipore, Darmstadt, Germany). Chromatin samples were immunoprecipitated with antibodies against a negative control normal mouse IgG, H3K9ac or Sirt6 (Abcam, Cambridge, MA). PCR amplification was performed in 20 μl volumes for 30–35 cycles to determine the appropriate conditions for the PCR products of each region. Primer sequences are shown in Supplementary Table 3. Antibodies used in this study are summarized in Supplementary Table 5.

**RNA extraction and real time RT-PCR**. Total RNA was isolated from mouse kidney or cells using TRIzol reagent (Invitrogen). The mRNA expression levels were determined by real-time quantitative RT-PCR using a Bio-Rad iCycler system (Bio-Rad, Hercules, CA).The sequences of specific primers are listed in Supplementary Table 4.

**Western blot and immunoprecipitation analyses**. Total cellular lysates preparation and western blot analysis were performed as follows: cells were collected with RIPA buffer (150 mM NaCl, 0.5% sodium deoxycholate, 0.1% SDS, 1% NP40, 1 mM EDTA and 50 mM Tris pH 8.0), followed by separation with sodium dodecyl sulfate–polyacrylamide gel electrophoresis. Then samples were transferred to PVDF membrane and incubated with primary antibody in 5% milk. Antibodies used in this study are summarized in Supplementary Table 5. To document the loading controls, the membrane was reprobed with a primary antibody against housekeeping protein GAPDH. The uncropped blots were shown in Supplementary Figs. 12 and 13.

**Immunofluorescence staining and confocal microscopy**. Immunofluorescent staining and images were obtained by a LSM780 laser scanning confocal microscope (ZEISS, Germany) system[50]. To monitor the various stages of autophagy, the tandem GFP-RFP-LC3 adenovirus construct obtained from Hanbio Inc (Shanghai, China) was used in this study.

**Microarray analysis**. The microarray experiments were performed by Shanghai Biotechnology Corporation (Shanghai, China). Total RNA was isolated using TRIzol reagent (Invitrogen) from three replicate samples of human podocytes. Agilent Whole Human Genome Oligo Microarray (4 × 44 K) (Agilent Technologies, Santa Clara, CA, USA) was used for transcriptome analysis. Microarray data was normalized using GeneSpring GX software (Agilent Technologies) and genes were categorized into multiple biological pathways using David database (https://david.ncifcrf.gov/).

**Statistics**. Data are expressed as means ± SE. Student's *t*-test was employed for comparisons between two groups; one-way analysis of variance (ANOVA) followed by Tukey's post-test for multiple comparisons was used for groups of three or more. All tests were two-tailed, and $P < 0.05$ was considered statistically significant.

**Data availability**. Microarray data sets have been deposited to Gene Expression Omnibus under accession code GSE100185. Other data that support the findings of this study are available from the corresponding author upon reasonable request.

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

## Acknowledgements

This study was supported by the National Science Fund for Distinguished Young Scholars to F.Y. (81525005); The National Nature Science Foundation of China (91642204, 81470958, 81371317, 81600570 and 81400730).

## Author contributions

M.L. and K.L. designed and conducted in vivo and in vitro experiments, performed data analysis, and helped write the manuscript. X.J.W. contributed to the experimental design and performed in vitro experiments on podocytes. M.Z. performed the confocal microscopy. Z.W., Y.Z., Z.Z. and X.W. performed in vivo animal studies. J.Z., H.S., C.Z. and J.P. analyzed human renal biopsy samples. Y.S. performed genotyping of mice. N.J., C.G. helped design experiments and interpreted the data. F.Y. designed the experiment, interpreted the data, wrote the manuscript, and approved the final version of the manuscript for publication.

## Additional information

**Competing interests:** The authors declare no competing financial interests.

