## [Peer Review file · Nature Communications]

Editorial Note: Parts of this peer review file have been redacted as indicated to remove third-party material where no permission to publish could be obtained

Reviewers' comments:

Reviewer #1 (Remarks to the Author):

This original investigation by Liu et al. proposes a role for sirtuin 6 in kidney podocytes in controlling Notch 1 and 4 by being a transcriptional repressor. The authors suggest that SIRT6 directly suppresses the expression of Notch1 and Notch4 by deacetylating H3K9 at their promoters. They further show that patients with glomerular diseases have decreased sirt 6 expression including patients from Minimal Change disease.

Several animal models are being used such as STZ diabetic mice and ADR treated mice (FSGS toxic model). Deletion of sirt6 aggravates injury and lentiviral over expression of sirt6 mitigates injury. Overall, this is a very well performed study, clearly relevant to kidney disease and potentially important for therapeutic considerations. The authors show rescue of sirt6 deficiency in podocytes by using a notch inhibitor.

The paper is rich in experimental data but I think a few more clarifying experiments are necessary.

1) MCD is a non-progressive renal disease and one would argue should not behave such as Membranous disease, diabetic nephropathy or FSGS. I am concerned that the finding of sirt6 down regulation in progressive and non-progressive disease casts some doubt on the validity or the relevance of sirt6 down modulation. One suggestion could be to take podocytes from the urine of MCD and FSGS patients and check for sirt6 down modulation but see if apoptosis signaling pathways are different.

2) Cultured podocytes are analyzed for expression of inflammatory mediators under HG conditions with high sirt6 expression. A bona fide podocyte injury molecule induced by conditions studied by the authors (FSGS, Lupus and Diabetic nephropathy) characterizing progressive renal disease is podocyte uPAR (Wei et al Nat Med 2008). The authors should perform uPAR expression profile under HG conditions and see if sirt6 over expression modulates uPAR expression

3) the authors abbreviate proteinic kidney disease with PKD which is commonly used as polycystic kidney disease acronym. This should be changed to not confuse broad readership.

Reviewer #2 (Remarks to the Author):

The manuscript by Liu et al., examines the role of the histone deacetylase SIRT6 in the progression of STZ-induced diabetic nephropathy and adriamycin induced nephropathy. Profiling of SIRT6 mRNA and protein expression obtained from patient biopsies, the authors noticed downregulation of SIRT6, which correlated with eGFR. The authors next deleted SIRT6 specifically from podocytes and observed worsening of kidney injury. In vitro studies in podocytes revealed a role for SIRT6 in inflammation, cytoskeletal remodeling, apoptosis and autophagy. Mechanistically the authors demonstrate that H3K9ac of NOTCH1/4 is modulated by SIRT6, leading to reduce NOTCH1/4 expression.

Overall the results of this study are interesting, but not unexpected. This study seems to be the first to investigate the role of SIRT6 specifically within podocytes in relation to kidney injury. Additionally, this is the first description of NOTCH1 and NOTCH4 as direct targets of SIRT6. However, several important issues need to be carefully addressed.

1. One major problem with the study is that the authors suggest that SIRT6 is bound to promoters of Notch1 and Notch4 genes, and overexpression of SIRT6 would reduce its binding to these promoters. If this is the case, it is unclear why overexpression of SIRT6 would reduce its binding to the promoters. What the results actually suggest is that H3K9ac is increased with high glucose, and reduced when SIRT6 is overexpressed in HG milieu. In this case, the experiment gives no indication for SIRT6 binding. To support their conclusions, the authors should provide ChIP assays

using a SIRT6 antibody, and not H3K9ac antibodies.

2. The link between Sirt6 and Notch1/4 signaling on autophagy is not experimentally established in this manuscript.

3. Since the lentiviral delivery does not seem to be specific to podocytes, the experiments involving gene delivery of Sirt6 to podocytes should be performed in Cre+;Sirt6^{fl/fl} as this would provide a better approach to answer whether Sirt6 is necessary and sufficient to prevent podocyte injury.

4. It seems that SIRT6 levels are reduced in the kidney and not specifically in podocytes. It is unclear whether SIRT6 expression is reduced only in the kidney or a global phenomenon (in other tissues) in high glucose conditions.

5. Despite the notion that other Sirtuins are important in a variety of kidney diseases, the rationale for investigating SIRT6 is not quite clear.

6. Does lentiviral mediated Sirt6 overexpression in vivo (Fig. 9) prevent ADR mediated H3K9 acetylation enrichment at Notch1 and 4 promoters?

7. Does Notch1 and Notch4 loss of function in vitro, either through siRNA or shRNA in the context of SIRT6 loss of function, block the pathogenic effect of Notch signaling on podocytes? Specifically, does loss of Notch1 and Notch4 rescue the SIRT6 loss of function phenotype?

8. Following overexpression of Sirt6 (Fig. 9), do markers of autophagy become altered in vivo? Figure 5E shows a very minimal difference. The data showing SIRT6 restores autophagy is not convincing.

9. Since Notch has multiple receptors and multiple ligands, which may play redundant roles in Notch signaling activation, can the authors demonstrate, either in vitro or in vivo, how Notch signaling is altered following depletion of Sirt6 or reintroduction of Sirt6?

10. Figure 9B should provide a higher magnification of the glomerulus or a zoomed-in inset. It seems that SIRT6 expression and overexpression is primarily cytoplasmic, the authors assert (and even show by IHC) that Sirt6 is one of the Sirtuins more commonly localized to the nucleus, where it deacetylates histones. Therefore, these immunofluorescence images are not quite convincing regarding the successful delivery of Sirt6 to the nucleus.

11. There is no quantification for mesangial matrix expansion in Fig. 3D, Fig. 4B and Fig. 8B.

12. The authors should also quantify glomerular basement membrane thickening for the EM experiments, especially in in Fig. 4C, as they suggest GBM thickening is enhanced in SIRT6 KO STZ mice.

13. The wording of the paper should be revised to more accurately describe the results. For instance on page 7 line 153, the findings don't necessarily suggest that SIRT6 will prevent podocyte injury as they don't overexpress SIRT6 in a diabetic model.

14. Page 8 line 181 the authors state they did a microarray to find Notch signaling being regulated by SIRT6 however the data are not shown. For a major point in the paper, the data should be added to the manuscript.

15. Figure 5C does not have the x-axis labeled and the text is too small to read. Also the cytometry read out plot (HG/AD-Ctrl - 10.36%) and the quantitation graph (HG/AD-Ctrl -14%) do not match up.

16. Figure 3B should have some sort of quantitation or Western to determine how much of a reduction in SIRT6 protein is achieved.

17. Figure 5A should include high glucose conditions (i.e. same conditions as Figure 6C) to see the levels or SIRT6 overexpression compared to wild-type when treated with high glucose. Also the figure needs to be cleaned up.

18. The type of STZ model used in this study is not a conventional model of diabetic nephropathy. The authors use a STZ-treated uninephrectomized model on high fat diet.

Reviewer #3 (Remarks to the Author):

The authors demonstrated the role of murine Sirt6 in podocytes in DKD and ADR models using their podocyte-specific Sirt6 CKO mice and lentiviral gene delivery systems. In addition, they assessed the role of human Sirt6 in various types of renal diseases using renal biopsy samples.

Some aspects of their findings seem to be novel as the role of Sirt6 in kidneys has not been previously reported. However, the authors should assess their data more carefully as highlighted below.

Major comments

1. Sirt6 is reported to localize in the nucleus. However, many figures submitted in this manuscript, such as Fig 1, 2, 3, and 9, show Sirt6 to be strongly stained in the cytoplasm. This appears inconsistent with previous reports. The authors should assess the intracellular localization of Sirt6 in various types of renal cells in vivo and in vitro.

2. What is the mechanism whereby Sirt6 was decreased in podocytes? The authors showed that both high glucose and ADR reduced Sirt6 levels in murine models. However, they did not explain the involved mechanisms. Please provide further information on what the most important stimulatory factor to downregulate Sirt6 expression in podocytes is.

3. In diabetic nephropathy models, the authors suggested that both AGE and high glucose were involved in decreasing Sirt6 levels. If this is the case, exposure would likely affect not only podocytes but also other types of cells, including mesangial cells, glomerular endothelial cells, parietal epithelial cells, and tubular cells. However, the downregulation of Sirt6 protein levels caused by diabetic kidney diseases was limited to podocytes. Please explain the mechanism because of which such effect was only observed in podocytes.

4. Please describe the background of the Sirt6 flox and Podocin cre mice which the authors used. Sirt6 flox provided by Jackson laboratory is based on FVB/N mice, while Podocin Cre is created with a C57/B6J background. Did you backcross before crossbreeding these flox-Cre mice to generate Sirt6 CKO mice?

5. The authors should evaluate the expression of other isoforms of sirtuins in Sirt6 CKO mice. For example, previous reports clarified that Sirt1 regulates Sirt6 protein expression (Kim HS et al. Hepatic-specific disruption of SIRT6 in mice results in fatty liver formation due to enhanced glycolysis and triglyceride synthesis. *Cell Metab.* 2010 Sep 8;12(3):224-36). There may be some possibilities for other sirtuins to compensate for the Sirt6 deficiency to protect against renal damages.

Minor comments

1. Page 8, line 180: the authors should display the microarray data results in this manuscript.

2. In many immunoblotting figures, some bands corresponding with Sirt6 appear to be double bands. An explanation on the possibility of post-translational modification of Sirt6 in these assays should be included in the manuscript.

3. Page 5, line 120 and 122: there are some typographical errors. For example, synaptopodin should be changed to synaptopodin. These problems require careful attention from the authors.

4. The authors should mention the sex of mice used in these experiments. For example, Yoshino J reported that Nicotinamide mononucleotide, a key NAD⁺ intermediate, can treat the pathophysiology of diet- and age-induced diabetes in mice (*Cell Metab.* 2011 Oct 5; 14(4): 528–536). In this paper, the effect of NMN and the activation of sirtuin were significantly different for male and female mice. Thus, the authors should analyze the effect of Sirt6 knockdown for both sexes.

Responses to Reviewers

We thank the reviewers for their thoughtful and constructive comments, which have guided the revision of this manuscript (Manuscript ID NCOMMS-16-18860R1). The changes in the manuscript have been highlighted in yellow for easy identification. The concerns have been addressed as follows:

To Reviewer #1:

1. MCD is a non-progressive renal disease and one would argue should not behave such as Membranous disease, diabetic nephropathy or FSGS. I am concerned that the finding of sirt6 down regulation in progressive and non-progressive disease casts some doubt on the validity or the relevance of sirt6 down modulation. One suggestion could be to take podocytes from the urine of MCD and FSGS patients and check for sirt6 down modulation but see if apoptosis signaling pathways are different.

Answer: We really appreciate the reviewer's question! Podocyte injury is integral to the pathogenesis of primary glomerulopathies, such as focal segmental glomerular sclerosis (FSGS), minimal change disease (MCD), and membranous nephropathy. Although the inciting injury to the podocyte may vary between these glomerular diseases, the inevitable consequence of podocyte injury is actin cytoskeleton derangement, apical redistribution or loss of slit diaphragm proteins, and loss of structural integrity, leading to eventual foot process effacement and podocyte detachment or apoptosis. However, as the reviewer pointed out, MCD is a non-progressive renal disease and histological findings of MCD in glomeruli are typically normal by light microscopy and only electron microscopy shows *effacement of podocyte foot processes* without electron-dense immune deposits. These manifestations are typically reversible with the use of corticosteroid therapy, so that progressive loss of renal function is rare.

The reviewer give us a very good suggestion for the detection of podocytes in urine. The assessment of podocyte injury can be accomplished by monitoring the number of podocyte cells in the urine. We collected urine samples and prepared cytospin slides from patients with MCD, membranous disease, FSGS, diabetic nephropathy, and normal individually as described (Fall B et al. Urinary Podocyte Loss Is Increased in Patients with Fabry Disease and Correlates with Clinical Severity of Fabry Nephropathy. PLoS One. 2016; 11(12):e0168346). By immunofluorescence staining with podocalyxin (a typical podocyte marker) in cytospin slides, it is very hard to obtain podocytes (podocalyxin-positive cells) in urine from normal individuals (only find 1-2 podocytes in one field of view with 20x objective). However, the number of urinary podocytes is much higher in patients with membranous disease, FSGS and diabetic nephropathy than in MCD patients (see the following figure).

In addition, we also performed immunofluorescence staining by colocalization of SIRT6 with podocalyxin. It seems that the intensity of green signal (SIRT6) were weak or not shown in most of podocalyxin-positive cells in patients with membranous nephropathy, FSGS or diabetic nephropathy, indicating the reduction of podocyte SIRT6 in these patients. However, it is hard

to identify the relative expression levels of SIRT6 in different diseases due to different amount of podocytes in urine from patients with different diseases. In particular, the amount of podocytes is very low in patients with MCD and among them, some are SIRT6-positive, some are not. Therefore, it is not very feasible to measure the expression level of SIRT6 in one single cell from urine. Collectively, based on our current studies and our available technique, it is difficult to get accurate conclusion regarding the reduction of SIRT6 and their apoptosis signaling difference between MCD and FSGS.

On the other hand, although in this study we found that the reduction of SIRT6 in renal biopsies from patients with different forms of podocytopathies including FSGS, MCD, membranous glomerulonephritis, IgA nephropathy and diabetic nephropathy, we can not exclude the different contributions of SIRT6 reduction to podocyte function in different diseases or different stage of disease, because SIRT6 had pleiotropic protective actions not only anti-apoptosis, but also anti-inflammation, maintenance of actin cytoskeleton. In this manuscript, we examined autophagy and apoptosis in podocytes with HG or ADR treatment, this may not be very suitable for MCD pathogenesis. To avoid misunderstanding and make our conclusion more accurately, we removed related data for MCD patients. But it is a very interesting topic and further studies need to investigate whether the reduction of SIRT6 contributes to inflammation, or actin cytoskeleton derangement, or other potential mechanisms in MCD.

Thanks again for your question!

Representative confocal microscopic images showing the colocalization of SIRT6 with podocalyxin in the urine from patients with different forms of podocytopathies.

2. Cultured podocytes are analyzed for expression of inflammatory mediators under HG conditions with high sirt6 expression. A bonafide podocyte injury molecule induced by conditions studied by the authors (FSGS, Lupus and Diabetic nephropathy) characterizing progressive renal disease is podocyte uPAR (Wei et al Nat Med 2008). The authors should perform uPAR expression profile under HG conditions and see if sirt6 over expression modulates uPAR expression.

Answer: Thank you for your thoughtful and constructive comments. Podocyte dysfunction, represented by foot process effacement and proteinuria, is often the starting point for progressive kidney disease. Studies have indicated that the induction of urokinase plasminogen activator receptor (uPAR) signaling in podocytes leads to foot process effacement and urinary protein loss in puromycin aminonucleoside (PAN) nephrosis and LPS-induced transient proteinuria in mice, which was also identified in a majority of patients with FSGS as a soluble factor that acts through binding to and activating podocyte $\beta 3$ integrin to promote cell motility and kidney permeability (Wei C, et al. Modification of kidney barrier function by the urokinase receptor. *Nat Med*. 2008;14(1):55-63); Wei C, et al. Circulating urokinase receptor as a cause of focal segmental glomerulosclerosis. *Nat Med*, 2011, 17(8); 952-960). In addition, studies have also shown that NFAT2 inhibitor ameliorates diabetic nephropathy and podocyte injury in db/db mice by inhibiting the expression of uPAR (Zhang L et al. NFAT2 inhibitor ameliorates diabetic nephropathy and podocyte injury in db/db mice. *Br J Pharmacol*. 2013;170(2):426-39). As suggested, in this study, we further detected the uPAR expression profile in different groups of mice and found the low-level of uPAR expression in podocytes from control mice. In contrast, diabetic Cre⁺/SIRT6^{+/+} mice had a significant increase in podocyte uPAR expression, an even stronger induction of uPAR expression was found in podocytes from diabetic Cre⁺/SIRT6^{fl/fl} mice (**Figure 4e**). In vitro, overexpression of SIRT6 also reduced the expression levels of uPAR in podocytes with HG or PAN treatment (**Figure S4**). Together, our results indicate that SIRT6 modulates uPAR expression. **The related results were described in page 7-8, discussed in page 13-14 highlighted in yellow and Figure 4 and S4.**

Representative confocal microscopic images showing the expressions of uPAR in the kidney from different groups of mice.

The expression level of urokinase plasminogen activator receptor (uPAR) in podocytes with puromycin aminonucleoside (PAN) and HG treatment, PAN as a positive control.

3. The authors abbreviate proteinic kidney disease with PKD which is commonly used as polycystic kidney disease acronym. This should be changed to not confuse broad readership.

Answer: Thank you for your careful review. We have checked carefully and corrected the abbreviations in this revision.

To Reviewer #2:

1. One major problem with the study is that the authors suggest that SIRT6 is bound to promoters of Notch1 and Notch4 genes, and overexpression of SIRT6 would reduce its binding to these promoters. If this is the case, it is unclear why overexpression of SIRT6 would reduce its binding to the promoters. What the results actually suggest is that H3K9ac is increased with high glucose, and reduced when SIRT6 is overexpressed in HG milieu. In this case, the experiment gives no indication for SIRT6 binding. To support their conclusions, the authors should provide ChIP assays using a SIRT6 antibody, and not H3K9ac antibodies.

Answer: Thank you very much for your very constructive comments. It is very important to make our conclusion safely. As suggested, our results revealed the increased levels of H3K9ac in the promoters of Notch1 and Notch4 in podocytes with HG treatment (Figure 6f). To confirm whether SIRT6 can recruit and deacetylate H3K9 on these promoters, we performed ChIP assay in podocytes with overexpression of SIRT6 and found that overexpression of SIRT6 significantly reduced the level of H3K9ac in the promoter region of Notch1 and Notch4 (Figure 6f). By ChIP assays using a SIRT6 antibody, our results further confirmed that SIRT6 could bind to the promoter of Notch1 and Notch4 (Figure 6g). Under HG condition, SIRT6 was dissociated from the promoters of these genes in podocytes and recruited to them after SIRT6-adenovirus transfection by ChIP analysis (Figure 6g). Collectively, we proposed that SIRT6 repressed the expression of Notch1 and Notch4 through the deacetylation of histones in their promoter regions. **The related results were added in page 9 highlighted in yellow and in Figure 6.**

2. The link between Sirt6 and Notch1/4 signaling on autophagy is not experimentally established in this manuscript.

Answer: Thank you for your question. To investigate the link between SIRT6 and Notch1/4 signaling on autophagy, gene silencing of SIRT6 and Notch1 was performed in this study. As well as gene silencing of Notch 1 restored the levels of autophagy-associated proteins in podocytes with HG treatment (Figure S6), gene silencing of Notch1 can rescue the autophagy defect in SIRT6-deficient podocytes as evidenced by the increased autophagy-associated protein levels (Figure 7e) and autophagy flux (Figure 7f). Similar results were also observed in SIRT6-deficient podocytes with gene silencing of Notch 4. **The related results were added in page 10 highlighted in yellow and in Figure 7 and S6.**

3. Since the lentiviral delivery does not seem to be specific to podocytes, the experiments involving gene delivery of Sirt6 to podocytes should be performed in Cre⁺/Sirt6^{fl/fl} as this would provide a better approach to answer whether Sirt6 is necessary and sufficient to prevent podocyte injury.

Answer: Thanks for your careful review and your suggestion. As suggested, recombinant lentivirus vectors pGLV3 harboring SIRT6 were also delivered into the mouse kidney by means of intraparenchymal injections in Cre⁺/Sirt6^{fl/fl} diabetic mice. Our results confirmed the

efficiency of in vivo gene transfer showing that mice received SIRT6 lentivirus markedly increased the levels of SIRT6 in isolated glomeruli by real time RT-PCR analysis (Figure 9a) and podocytes by immunofluorescent analysis after 5-week injection (Figure 9b). In vivo overexpression of SIRT6 significantly ameliorated renal injury as evidenced by reduced albuminuria (Figure 9c), decreased mesangial expansion (Figure 9d) and ameliorated podocyte injury in Cre⁺/SIRT6^{fl/fl} diabetic mice (Figure 9e). Overexpression of SIRT6 recovered nephrin and podocin expression (Figure 9f) and increased the levels of LC3 and beclin1 by IHC staining (Figure S8). Similar results were also found in wild-type mice with overexpression of SIRT6 by intrarenal lentiviral gene delivery in ADR nephropathy.

In addition, we finally measured the levels of Notch1, Notch4 and H3K9ac in the kidney after transfection with SIRT6 lentivirus in Cre⁺/SIRT6^{fl/fl} ADR-treated mice. It was found that the levels of Notch1 (Figure 10a), Notch4 (Figure 10b) and H3K9ac (Figure 10c) were significantly decreased in podocytes after transfection with SIRT6 lentivirus in Cre⁺/SIRT6^{fl/fl} ADR-treated mice. ChIP assay in isolated glomeruli further confirmed that overexpression of SIRT6 directly suppresses the expression of Notch1 and Notch4 by deacetylating H3K9 at their promoters (Figure 10d-e). **The related results were added in page 11 highlighted in yellow and Figure 9 and 10.**

4. It seems that SIRT6 levels are reduced in the kidney and not specifically in podocytes. It is unclear whether SIRT6 expression is reduced only in the kidney or a global phenomenon (in other tissues) in high glucose conditions.

Answer: Thank you for your question. As suggested, we performed the SIRT6 expression profile in different tissues including heart, brain, liver, spleen and lung in diabetic mice. It was found that besides the kidney, the expression level of SIRT6 was also reduced in the heart and lung in diabetic mice, there was no significant changes in the brain, spleen and liver (the following figure).

In addition, we have also characterized the expression of SIRT6 in other renal parenchymal cells such as rat glomerular mesangial cells, rat glomerular endothelial cells, and human tubule epithelial cells (HK-2) and found that they had different expression patterns of SIRT6 in response to hyperglycemia (Figure S2) or ADR (Figure S3), suggesting that SIRT6 may serve different functions in different tissues or cells. **The related results were added in page 6 highlighted in yellow and Figure S2 and S3.**

5. Despite the notion that other Sirtuins are important in a variety of kidney diseases, the rationale for investigating SIRT6 is not quite clear.

Answer: Thank you very much for your question! We have three reasons to study the role of SIRT6 in podocytes:

1. In fact, we firstly assessed the expression patterns of SIRT6 in the kidney from STZ-induced diabetic mice and ADR-treated mice (a single injection of ADR leads to significant glomerular damage that recapitulates the human disease of FSGS) individually, two independent models for proteinuric kidney disease. Our results showed that the levels of SIRT1, SIRT3, SIRT4 and SIRT6 were reduced in the kidney from STZ-induced diabetic mice. In ADR nephropathy, SIRT1, SIRT5 and SIRT6 were downregulated in the kidney from ADR-treated mice, but the expression SIRT4 was shown an increase tendency, indicating different expression patterns of SIRTs under different pathological conditions (Figure 1a). Considering that both SIRT1 and SIRT6 were reduced in diabetic nephropathy and ADR nephropathy and recent studies have highlighted the contribution of SIRT1 to the regulation of renal function (Kazuhiro Hasegawa, et al. Renal tubular Sirt1 attenuates diabetic albuminuria by epigenetically suppressing Claudin-1 overexpression in podocytes. *Nat Med*, 2013, 19(11), 1496-1504; Ruijie Liu, et al. Role of Transcription Factor Acetylation in Diabetic Kidney Disease. *Diabetes*, 2014, 63(7), 2440-

2453), the present study was to further explore the role of SIRT6 in the kidney.

2. More importantly, we observed the reduction of SIRT6 in the kidney from patients with podocytopathies including FSGS, diabetic nephropathy, MGN, and IgA nephropathy. Furthermore, we found a positive correlation between SIRT6 expression and estimated glomerular filtration rate and a negative correlation between SIRT6 and proteinuria in all subjects, suggesting that the reduction of SIRT6 may be a common feature of human proteinuric kidney disease.

3. In vitro, our results also revealed that high glucose or AGE significantly reduced podocyte SIRT6 expression in a concentration dependent manner. In other renal parenchymal cells, SIRT6 expression was not changed in rat glomerular mesangial cells and human tubule epithelial cells HK-2, although a decrease tendency of SIRT6 expression was observed in rat glomerular endothelial cells under HG condition. In addition, we also demonstrated the reduction of SIRT6 expression in ADR-treated podocytes, rat glomerular endothelial cells and human tubule epithelial cells HK-2, no significant changes in rat glomerular mesangial cells. Collectively, these findings lead us to speculate that renal SIRT6 may have different expression patterns depending on cell type or environmental stimuli, but SIRT6 has the common response in podocytes with HG, AGE or ADR treatment.

Therefore, we explore the role of SIRT6 on the regulation of podocyte function by podocyte-specific SIRT6 knockout ($Cre^+/SIRT6^{fl/fl}$) mice. **The related results were added in page 5, discussed in page 12 highlighted in yellow and Figure 1.**

6. Does lentiviral mediated Sirt6 overexpression in vivo (Fig. 9) prevent ADR mediated H3K9 acetylation enrichment at Notch1 and 4 promoters?

Answer: We further measured the levels of Notch1, Notch4 and H3K9ac in the kidney after transfection with SIRT6 lentivirus in $Cre^+/SIRT6^{fl/fl}$ ADR-treated mice. It was found that the levels of Notch1 (Figure 10a), Notch4 (Figure 10b) and H3K9ac (Figure 10c) were significantly decreased in podocytes after transfection with SIRT6 lentivirus in $Cre^+/SIRT6^{fl/fl}$ ADR-treated mice. ChIP assay in isolated glomeruli further confirmed that overexpression of SIRT6 directly suppresses the expression of Notch1 and Notch4 by deacetylating H3K9 at their promoters (Figure 10d-e). **The related results were added in page 11 highlighted in yellow and Figure 10.**

7. Does Notch1 and Notch4 loss of function in vitro, either through siRNA or shRNA in the context of SIRT6 loss of function, block the pathogenic effect of Notch signaling on podocytes? Specifically, does loss of Notch1 and Notch4 rescue the SIRT6 loss of function phenotype?

Answer: Thanks for your careful review and constructive suggestion. Inhibition of Notch signaling reduced inflammatory responses (Figure 7a) and apoptosis (Figure 7b), ameliorated actin cytoskeleton derangement (Figure 7c), and recovered the expression levels of nephrin and podocin (Figure 7d) in SIRT6-deficient podocytes. As well as gene silencing of Notch 1 restored the levels of autophagy-associated proteins in podocytes with HG treatment (Figure S6), gene silencing of Notch1 can rescue the autophagy defect in SIRT6-deficient podocytes

as evidenced by the increased the autophagy-associated protein levels (Figure 7e) and autophagy flux (Figure 7f). Similar results were also observed in SIRT6-deficient podocytes with gene silencing of Notch 4 (data not shown). **The related results were added in page 10 highlighted in yellow and Figure 7 and S6.**

8. Following overexpression of Sirt6 (Fig. 9), do markers of autophagy become altered in vivo? Figure 5E shows a very minimal difference. The data showing SIRT6 restores autophagy is not convincing.

Answer: Thanks for your careful review. To answer the reviewer's question, we measured the marker of autophagy in vivo. We measured the expression levels of autophagy-associated proteins by IHC staining, it was found that over expression of SIRT6 restored the levels of restores autophagy. Furthermore, we redid the experiment related experiments and further detected other autophagy-associated proteins such as ATG5 and ATG12 in vitro and confirmed the changes in autophagy after overexpression of SIRT6. **The related results were added in page 8 and 11 highlighted in yellow and Figure 5 and S8.**

9. Since Notch has multiple receptors and multiple ligands, which may play redundant roles in Notch signaling activation, can the authors demonstrate, either in vitro or in vivo, how Notch signaling is altered following depletion of Sirt6 or reintroduction of Sirt6?

Answer: Thanks for your question. Although we did not attempt to explore how Notch signaling is modulated by SIRT6, our results may provide some information.

1. The Notch pathway consists of four Notch receptors (Notch1-4) and five Notch ligands including Jagged1, Jagged2, Delta-like (Dll) 1, Dll3, and Dll4. Although some ligands are upregulated in response to HG in podocytes, our microarray results did not find significant changed of Notch ligand in response to SIRT6 overexpression, indicating that SIRT6 has no effects on the levels of ligands.

2. We measured the levels of H3K9 acetylation (H3K9ac) in ADR nephropathy. It was found that the H3K9ac levels were significantly increased in the renal biopsies and mice with ADR nephropathy mice by IHC staining, which were ameliorated by overexpression of SIRT6. In vitro, our results revealed the increased levels of H3K9ac in the promoters of Notch1 and Notch4 in podocytes with HG treatment (Figure 6f). To confirm whether SIRT6 can recruit and deacetylate H3K9 on these promoters, we performed ChIP assay after overexpression of SIRT6 in podocytes under HG condition and found that overexpression of SIRT6 significantly reduced the level of H3K9ac in the promoter region of Notch1 and Notch4 (Figure 6f). We further investigated whether SIRT6 could bind to the promoter of Notch1 and Notch4. ChIP analysis showed that SIRT6 could bind to the promoter of Notch1 and Notch4, and SIRT6 was dissociated from the promoters of these genes in podocytes with HG treatment and recruited to them after SIRT6-adenovirus transfection (Figure 6g). Therefore, our studies indicated that, by binding to the promoters of Notch1 and Notch 4 and deacetylating H3K9 at these sites, SIRT6 reduced the expression of Notch1 and Notch4. Further studies are needed to clarify the precise mechanisms of Notch signaling regulation

10. Figure 9B should provide a higher magnification of the glomerulus or a zoomed-in inset. It seems that SIRT6 expression and overexpression is primarily cytoplasmic, the authors assert (and even show by IHC) that Sirt6 is one of the Sirtuins more commonly localized to the nucleus, where it deacetylates histones. Therefore, these immunofluorescence images are not quite convincing regarding the successful delivery of Sirt6 to the nucleus.

Answer: Thanks for your careful review and your question! In fact, in vivo overexpression of SIRT6 by intrarenal lentiviral gene delivery in Cre⁺/SIRT6^{fl/fl} mice, we found that SIRT6 expression and overexpression is not only cytoplasm, but also in nucleus as shown in new Figure 9a with a higher magnification of the glomerulus. In addition, we re-performed related experiments and still found that SIRT6 was in the nucleus and the cytoplasm of some renal cells. Therefore, we further determined the intracellular localization of SIRT6 in different renal cells. Although SIRT6 was originally identified as a nuclear-localizing protein (Finkel T, Deng CX, Mostoslavsky R. Recent progress in the biology and physiology of sirtuins. *Nature* 2009, 460, 587-591), our current studies have indicated that its subcellular localization depends on cell type in the kidney. Except that SIRT6 localized in the nucleus of podocytes, SIRT6 were also found both in the nucleus and the cytoplasm of rat glomerular endothelial cells, rat glomerular mesangial cells and human proximal tubule epithelial cells by different assays (**Figure S9**). These findings lead us to speculate that renal SIRT6 may shuttle between the nucleus and cytoplasm, depending on cell type or environmental stimuli. **The related results were added in page 12-13 highlighted in yellow and Figure S9.**

Nuclear isolation assay and immunofluorescence staining was used to detect the intracellular localization of SIRT6 in various types of renal cells. (a) Representative Western blot gel documents and summarized data showing the intracellular localization of SIRT6 in various types of renal cells including human podocytes (HPC), rat glomerular endothelial cells (GENC), rat glomerular mesangial cells (MC), and human proximal tubule epithelial cells (HK-2). **(b)** Representative confocal microscopic images showing the intracellular localization of SIRT6 in various types of renal cells.

11. There is no quantification for mesangial matrix expansion in Fig. 3D, Fig. 4B and Fig. 8B.

Answer: Thanks for your careful suggestion. According to previous studies, we have quantified mesangial matrix expansion in this revision. Light microscopic views of 40 consecutive glomerular cross-sections per mouse were scanned into a computer. Glomerular and mesangial matrix areas were quantified in a blinded fashion using an image analysis system (MetaMorph version 6.1; Universal Imaging). Mesangial matrix index (MMI) was calculated as the ratio of mesangial area to glomerular area *100 (% area) (Vasantha Kolavennu, et al. Targeting of RhoA/ROCK Signaling Ameliorates Progression of Diabetic Nephropathy Independent of Glucose Control. *Diabetes*. 2008, 57(3), 714-23). Glomerular injury was determined from kidney cross-sections after periodic acid Schiff staining using a semiquantitative score (0, no sclerosis; 1, mild segmental sclerosis; 2, moderate segmental sclerosis; 3, severe segmental sclerosis; and 4, global sclerosis) as previously described (Achim Lothar, et al. Deoxycorticosterone Acetate/Salt-Induced Cardiac But Not Renal Injury Is Mediated By

Endothelial Mineralocorticoid Receptors Independently From Blood Pressure. *Hypertension*. 2016, 67(1), 130-138). **The related results were added in related figures.**

12. The authors should also quantify glomerular basement membrane thickening for the EM experiments, especially in in Fig. 4C, as they suggest GBM thickening is enhanced in SIRT6 KO STZ mice.

Answer: Thanks for your suggestion. As suggested, we added the quantifications into this manuscript. Five glomeruli were randomly selected from each mouse and 10 transmission electron micrographs were taken in each glomerulus. We analyzed all the images by quantified the glomerular basement membrane (GBM) thickness, foot process width and the number of foot processes per μm of GBM as described in our previous studies (Fu Y, Sun Y, Zhou M, Wang X, Wang Z, Wei X, Zhang Y, Su Z, Liang K, Tang W, Yi F. Therapeutic Potential of Progranulin in Hyperhomocysteinemia-Induced Cardiorenal Dysfunction. *Hypertension*. 2016 Nov 21. pii: HYPERTENSIONAHA.116.08154. [Epub ahead of print]). **The related results were added in related figures.**

13. The wording of the paper should be revised to more accurately describe the results. For instance on page 7 line 153, the findings don't necessarily suggest that SIRT6 will prevent podocyte injury as they don't overexpress SIRT6 in a diabetic model.

Answer: Thank you for your careful review. As suggested, we reorganized our manuscript and removed the inaccurate description in this revision.

14. Page 8 line 181 the authors state they did a microarray to find Notch signaling being regulated by SIRT6 however the data are not shown. For a major point in the paper, the data should be added to the manuscript.

Answer: As suggested, we added the microarray results related our current findings in HG-treated podocytes with or without SIRT6 overexpression by Agilent Whole Human Genome Oligo Microarray for global gene expression analysis (Figure 6d, Representative heatmap of gene expression levels by multiplex qRT-PCR array). **The related results were added in page 9 highlighted in yellow and Figure 6.**

15. Figure 5C does not have the x-axis labeled and the text is too small to read. Also the cytometry read out plot (HG/AD-Ctrl – 10.36%) and the quantitation graph (HG/AD-Ctrl -14%) do not match up.

Answer: Thanks for your careful review. As suggested, we have added x-axis labeled and adjusted the size of figure. In the graph, Q1-LR represented early apoptosis and Q1-UR represented late apoptosis. Therefore, the quantitation graph included Q1-LR (early apoptosis) and Q1-UR (late apoptosis). **The related figures was revised in Figure 5.**

16. Figure 3B should have some sort of quantitation or Western to determine how much

of a reduction in SIRT6 protein is achieved.

Answer: Thanks for your careful review. As suggested, we added the quantifications into this manuscript. SIRT6 staining in the podocytes was quantified as the ratio of SIRT6⁺synaptopodin⁺ cells to synaptopodin⁺ cells using ImageJ 1.26t software according to previous study (Sandeep K. Mallipattu, et al. Krüppel-like factor 6 regulates mitochondrial function in the kidney. *J Clin Invest.* 2015, 125(3), 1347-13616). Additionally, we isolated glomeruli by using spherical Dynabeads to confirm the SIRT6 decrease in glomeruli from Cre⁺/SIRT6^{fl/fl} mice. The procedure of mice glomeruli isolation was based on previous studies (Takemoto M, et al. A new method for large scale isolation of kidney glomeruli from mice. *Am J Pathol.* 2002; 161(3):799-805; Liu X, et al. Isolating glomeruli from mice: A practical approach for beginners. *Exp Ther Med.* 2013 May; 5(5):1322-1326). Our data showed that the purity of glomeruli was estimated to be 98% (Representative isolated glomeruli were shown in the following figure). As shown in Figure 3c, the levels of SIRT6 were significantly reduced in isolated glomeruli from Cre⁺/SIRT6^{fl/fl} mice. **The related results were added in page 6 highlighted in yellow and Figure 3.**

17. Figure 5A should include high glucose conditions (i.e. same conditions as Figure 6C) to see the levels or SIRT6 overexpression compared to wild-type when treated with high glucose. Also the figure needs to be cleaned up.

Answer: Thanks for your careful review. We re-did this experiments including the groups of overexpression of SIRT6 by a SIRT6-adenovirus transfection in podocyte with or without high glucose, our results showed that overexpression of SIRT6 could restored the decrease of SIRT6 in high glucose. **The related figures was revised in Figure 5.**

18. The type of STZ model used in this study is not a conventional model of diabetic nephropathy. The authors use a STZ-treated uninephrectomized model on high fat diet.

Answer: Thanks for your careful review. C57BL/6 mice are relatively resistant to nephropathy after induction of diabetes by STZ (Brosius FC 3rd, et al. Mouse models of diabetic nephropathy. *J Am Soc Nephrol.* 2009, 20, 2503–251; Breyer MD, et al. Mouse models of diabetic nephropathy. *J Am Soc Nephrol.* 2005, 16, 27–45), and HFD provides a commonly used approach to induce obesity and insulin resistance in C57BL/6 strain, in particular, after the onset of metabolic syndrome, and mice on a HFD develop increased albuminuria and glomerular lesions (Soler MJ, et al. New experimental models of diabetic nephropathy in mice models of type 2 diabetes: efforts to replicate human nephropathy. *Exp Diabetes Res.* 2012, 2012: 616313), Therefore, we used a STZ-treated uninephrectomized mice model on a HFD to hasten the development of DN following previous studies. To induce diabetic nephropathy, male mice (10 weeks of age) were injected STZ (80 mg per kg body weight intraperitoneally daily for three days). (Kuwabara T, et al. Exacerbation of diabetic nephropathy by hyperlipidaemia is mediated by Toll-like receptor 4 in mice. *Diabetologia*, 2012, 55, 2256–2266). We have also successfully used this methods to induced diabetic nephropathy (Du P, *et al.* NOD2 promotes renal injury by exacerbating inflammation and podocyte insulin resistance in diabetic nephropathy. *Kidney International* | 2013, **84**, 265-276).

To Reviewer #3:

1. Sirt6 is reported to localize in the nucleus. However, many figures submitted in this manuscript, such as Fig 1, 2, 3, and 9, show Sirt6 to be strongly stained in the cytoplasm. This appears inconsistent with previous reports. The authors should assess the intracellular localization of Sirt6 in various types of renal cells.

Answer: Thanks for your careful review. We used immunofluorescence staining and nuclear isolation assay to detect the intracellular localization of SIRT6 in various types of renal cells in this study. Firstly, we characterized the expression profile of SIRT6 in various types of renal parenchymal cells such as rat glomerular mesangial cells, rat glomerular endothelial cells, human podocytes, and human proximal tubule epithelial cells (HK-2). It was found that SIRT6 was all expressed in these major types of renal cells. We further determined the intracellular localization of SIRT6 in these cells. Although SIRT6 was originally identified as a nuclear-localizing protein (Eriko Michishita, et al. Evolutionarily Conserved and Nonconserved Cellular Localizations and Functions of Human SIRT Proteins. *Mol Biol Cell*. 2005, 16(10), 4623-4635), our current studies have indicated that its subcellular localization depends on cell type in the kidney. Except that SIRT6 localized in the nucleus of podocytes, SIRT6 were also found both in the nucleus and the cytoplasm of rat glomerular endothelial cells, rat glomerular mesangial cells and human proximal tubule epithelial cells (Figure S9). These findings lead us to speculate that renal SIRT6 may shuttle between the nucleus and cytoplasm, depending on cell type or environmental stimuli. **The related results were added in page 12-13 highlighted in yellow and Figure S9.**

Nuclear isolation assay and immunofluorescence staining was used to detect the intracellular localization of SIRT6 in various types of renal cells. (a) Representative Western blot gel documents and summarized data showing the intracellular localization of SIRT6 in various types of renal cells including human podocytes (HPC), rat glomerular endothelial cells (GENC), rat glomerular mesangial cells (MC), and human proximal tubule epithelial cells (HK-2). **(b)** Representative confocal microscopic images showing the intracellular localization of SIRT6 in various types of renal cells.

2. What is the mechanism whereby Sirt6 was decreased in podocytes? The authors showed that both high glucose and ADR reduced Sirt6 levels in murine models. However, they did not explain the involved mechanisms. Please provide further information on what the most important stimulatory factor to downregulate Sirt6 expression in podocytes is.

Answer: Thank you very much for your question! Although we did not attempt to explore molecular mechanisms of SIRT6 regulation in podocytes under pathological conditions in this study, recent studies have reported that SIRT6 is regulated by post-transcriptionally at the RNA and protein levels. MiR-34a and -122 reduce SIRT6 mRNA stability. The MDM2 ubiquitylate can target SIRT6 protein for protease-dependent degradation. SIRT6 gene expression is also modulated by several transcription factors such as c-Fos, E2 transcription factor 1 (E2F1) and Runt-related transcription factor 2 (RUNX2). In addition, SIRT1 can regulate SIRT6 by forming a complex with FOXO3a and NRF1 on the promoter of SIRT6. In fact, our preliminary studies also found that the downregulation of Foxo3a and NRF1 in podocytes with HG treatment (Figure S10). Further studies are needed to clarify the precise mechanisms of SIRT6 regulation. **The related discussion was added in page 16 highlighted in yellow and Figure S10.**

3. In diabetic nephropathy models, the authors suggested that both AGE and high glucose were involved in decreasing Sirt6 levels. If this is the case, exposure would likely affect not only podocytes but also other types of cells, including mesangial cells, glomerular endothelial cells, parietal epithelial cells, and tubular cells. However, the downregulation of Sirt6 protein levels caused by diabetic kidney diseases was limited to podocytes. Please explain the mechanism because of which such effect was only observed in podocytes.

Answer: Thank you very much for your question!

1. In fact, we firstly assessed the expression patterns of SIRTs in the kidney from STZ-induced diabetic mice and ADR-treated mice (a single injection of ADR leads to significant glomerular damage that recapitulates the human disease of FSGS) individually, two independent models for proteinuric kidney disease. Our results showed that the levels of SIRT1, SIRT3, SIRT4 and SIRT6 were reduced in the kidney from STZ-induced diabetic mice. In ADR nephropathy, SIRT1, SIRT5 and SIRT6 were downregulated in the kidney from ADR-treated mice, but the expression SIRT4 was shown an increase tendency, indicating different expression patterns of SIRTs under different pathological conditions (Figure 1a).

2. In vitro, our results revealed that high glucose or AGE significantly reduced podocyte SIRT6 expression in a concentration dependent manner. In other renal parenchymal cells, SIRT6 expression was not changed in rat glomerular masangial cells and human tubule

epithelial cells HK-2, although a decrease tendency of SIRT6 expression was observed in rat glomerular endothelial cells under **the same HG condition**. In addition, we also demonstrated the reduction of SIRT6 expression in ADR-treated podocytes, rat glomerular endothelial cells and human tubule epithelial cells HK-2, no significant changes in rat glomerular mesangial cells. Considering that SIRT6 was reduced in podocyte in responses to all the stimuli including HG, AGE and ADR, the reduction of SIRT6 was also confirmed other different forms of podocytopathies such as membranous glomerulonephritis and IgA nephropathy treatment. Therefore, we focus on the role of SIRT6 in podocytes. Of course, we can not excluded the changes of SIRT6 in other renal cells under pathological conditions, although in vitro the same HG condition has only a big effect on podocyte SIRT6 expression. These findings lead us to speculate that renal SIRT6 may have different expression patterns depending on cell type, environmental stimuli and different sensitivity to pathological factors.

4. Please describe the background of the Sirt6 flox and Podocin cre mice which the authors used. Sirt6 flox provided by Jackson laboratory is based on FVB/N mice, while Podocin Cre is created with a C57/B6J background. Did you backcross before crossbreeding these flox-Cre mice to generate Sirt6 CKO mice?

Answer: Thanks for your question! We are sorry that we have not provide the related information before. Floxed SIRT6 mice with FVB/N background ($SIRT6^{flox/flox}$, Jackson Laboratory, Bar Harbor, ME) were backcrossed with C57BL/6 mice more than 12 generations to produce congenic strains. Then C57BL/6 $SIRT6^{flox/flox}$ mice were crossed with mice expressing Cre recombinase (Cre) under the control of the podocin promoter (B6.Cg-Tg [NPHS2-cre] 295Lbh/J; Jackson Laboratory) to generate podocyte-specific SIRT6 knockout mice (Podocin-Cre $SIRT6^{fl/fl}$ mice; $Cre^+/SIRT6^{fl/fl}$ mice). Mice with two WT alleles and Cre expression were used as controls (Podocin-Cre $SIRT6^{+/+}$; $Cre^+/SIRT6^{+/+}$). Genotyping by tail preparation and PCR were performed at 2 weeks of age. **The related information was added in page 17 highlighted in yellow.**

5. The authors should evaluate the expression of other isoforms of sirtuins in Sirt6 CKO mice. For example, previous reports clarified that Sirt1 regulates Sirt6 protein expression (Kim HS et al. Hepatic-specific disruption of SIRT6 in mice results in fatty liver formation due to enhanced glycolysis and triglyceride synthesis. Cell Metab. 2010 Sep 8;12(3):224-

36). There may be some possibilities for other sirtuins to compensate for the Sirt6 deficiency to protect against renal damages.

Answer: Thank you for your suggestion! We also evaluated the expression of other isoforms of sirtuins in glomeruli from Cre⁺/SIRT6^{fl/fl} mice in vivo or SIRT6-deficient podocytes in vitro. At the expression levels, we did not find other sirtuins to compensate for the SIRT6 deficiency (Figure S11). **The related results were discussed in page 16 highlighted in yellow and Figure S11.**

Minor comments

1. Page 8, line 180: the authors should display the microarray data results in this manuscript.

Answer: Thanks for your careful review. As suggested, we added the microarray results related our current findings in HG-treated podocytes with or without SIRT6 overexpression by Agilent Whole Human Genome Oligo Microarray for global gene expression analysis (Figure 6d, Representative heatmap of gene expression levels by multiplex qRT-PCR array). **The related results were added in page 9 highlighted in yellow and Figure 6d.**

2. In many immunoblotting figures, some bands corresponding with Sirt6 appear to be double bands. An explanation on the possibility of post-translational modification of Sirt6 in these assays should be included in the manuscript.

Answer: We really appreciate your question and concerns! In fact, several possibilities may contribute to the double bands such as antibody specificity, cell type or post-translational modification. In our experiments, some sample showed double-band, some sample showed one-band. To further verify the observation, we bought another catalogue anti-SIRT6-antibody (ab62739) from abcam (Cambridge, MA) for Western blotting instead of the antibody we used before (anti-SIRT6 antibody, PB0375, Boster, Wuhan, China). As shown in our results, there is a single band in immunoblotting for SIRT6. Therefore, double bands possibly due to the specificity of antibody or cell type. To avoid misunderstanding, we re-performed the Western blot experiments for the detection of SIRT6 and replaced related some figures.

We used anti-SIRT6 antibody(PB0375) from Boster before, the sample in the data sheet is

[Redacted]

Our representative Western blot gel documents (the original blot for Figure 8D) are:

All lanes: Anti-SIRT6 antibody (PB0375) at 1/500 dilution

Lane 1-2: Human podocyte lysate

Lane 3-4: Human podocyte lysate (with ADR treatment).

Representative Western blot gel documents and summarized data showing the relative protein levels of SIRT6 in podocytes with ADR (ADR, 0.4 μ g/ml) for 24 hours.

In this revision, we bought another catalogue anti-SIRT6-antibody (ab62739) from abcam, the sample in the data sheet is:

[Redacted]

Our representative Western blot gel documents (the original blot for Figure 8D now moved to the new Figure 1f in this revision) are:

Representative Western blot gel documents and summarized data showing the relative protein levels of SIRT6 in podocytes with ADR (ADR, 0.4 μ g/ml) for 24 hours.

3. Page 5, line 120 and 122: there are some typographical errors. For example, synaptopotin should be changed to synaptopodin. These problems require careful attention from the authors.

Answer: We are so sorry for the spelling errors. We have checked carefully and corrected the spelling mistakes in this revision.

4. The authors should mention the sex of mice used in these experiments. For example, Yoshino J reported that Nicotinamide mononucleotide, a key NAD⁺ intermediate, can treat the pathophysiology of diet- and age-induced diabetes in mice (Cell Metab. 2011 Oct 5; 14(4): 528–536). In this paper, the effect of NMN and the activation of sirtuin were significantly different for male and female mice. Thus, the authors should analyze the effect of Sirt6 knockdown for both sexes.

Answer: We are so sorry that we have not provide the related information before. In fact, we used male mice in all experiments. To explore the effect of SIRT6 deficiency for both sexes, we observed the podocyte injury in male or female Cre⁺/SIRT6^{fl/fl} mice. As shown in the following table, there is no significant changes on blood glucose between male and female in diabetic mice, as well as podocyte differentiation markers including nephrin and podocin. Although Jun Yoshino, et al. had reported the effect of Nicotinamide mononucleotide (NMN) were significantly different for male and female HFD-induced T2D mice, NMN was administered for whole body. In our study, our mice were used is podocyte-specific loss of SIRT6 mice and podocyte had no obvious function on the blood glucose metabolism. Therefore, it is possible that podocyte-specific loss of SIRT6 had no different effect on podocytes between male and female mice.

Variable	Group	Sex	Glucose (mmol/l)
Cre ⁺ /SIRT6 ^{+/+}	Sham	male	6.1 ± 0.57
		female	5.9 ± 1.18
	STZ	male	19.7 ± 2.71*
		female	18.9 ± 3.08*
Cre ⁺ /SIRT6 ^{fl/fl}	Sham	male	6.3 ± 0.79
		female	6.2 ± 1.11
	STZ	male	20.3 ± 4.08*
		female	21.9 ± 3.57*

Reviewers' comments:

Reviewer #1 (Remarks to the Author):

The paper is much improved, the conclusions clear.

The new finding that Sirtuin6 mitigates podocyte uPAR expression is convincing that the author's mechanism is relevant to human glomerular disease.

Given the importance of this observation, i would ask for this finding to be briefly mentioned in the abstract.

Reviewer #2 (Remarks to the Author):

I carefully reviewed the author's response to my original comments. I am happy to report that I was impressed by their response and I believe that the manuscript is ready to go.

Reviewer #3 (Remarks to the Author):

The authors have demonstrated the role of murine Sirt6 in podocytes in DKD and ADR models using podocyte-specific Sirt6 CKO mice and lentiviral gene delivery systems. In addition, they have assessed the role of human Sirt6 in various types of renal diseases using renal biopsy samples. Because the authors have performed many additional experiments according to the comments provided by three reviewers, the manuscript seems to have improved when compared to previous versions. However, the authors should assess the following data more carefully. I am still concerned about the quality of immunoblots of seven isoforms of Sirt and immunostaining of Sirt6.

Major Comments:

1. In Figure 1c, immunofluorescence has revealed that positive areas of immunostaining of Sirt6 were co-localized with synaptopodin. This seems inconsistent with the authors' reply to Major Comment 1 raised by Reviewer 3. The authors have written that Sirt6 was localized in the nucleus of the podocytes. However, synaptopodin was localized in the cytoplasm and not in the nucleus. This result has led us to infer that Sirt6 was also localized in the cytoplasm to some extent as shown in other types of cells. Please present the necessary immunofluorescence data that can confirm the authors' answer to our comment.
2. In Figure 1a, the authors have presented the immunoblots of seven isoforms of Sirt. Left panels demonstrate the results showing the expression levels in the sham group and STZ treatment group. Right panels demonstrate the results of the sham group and ADR administration group. The intensities of the bands corresponding to Sirt2, Sirt6, and Sirt7 in the sham groups demonstrated in the left panels seem very different from those demonstrated in the right panels. Because the loading dose of proteins in western blots and antibodies used are the same between both the panels, the immunoblot results should also be similar. However, the intensities of the bands were totally different. Please repeat the immunoblots.
3. In Figure 1b, the authors have presented representative immunoblots of Sirt6 in upper panels and quantification of their intensities in lower graphs. However, the results seem incompatible. For example, the levels of Sirt6 in db/db are approximately 40% as compared to the levels in db/+. However, the upper figures manifest either no bands or very slight bands for Sirt6 in db/db. Please explain the inconsistency between the intensities of the bands in the images and their quantification.
4. In Figure 5a, the authors have presented immunoblots of Sirt6 in left panels and their quantification in right graph. The intensities of the bands for Sirt6 adenovirus vector seem several hundred times higher than those for control adenovirus vector. However, the right graph illustrates that the expression of Sirt6 in adenoviral vector transfection was about four times higher than the expression of Sirt6 transfected in control vector. Please analyze the data more carefully.

5. As a reply to Major Comment from Reviewer 3, the authors have explained that other types of renal cells, such as glomerular endothelial cells, mesangial cells, and proximal tubules, showed both nuclear and cytoplasmic Sirt6, which indicated the possibility of nuclear translocation of Sirt6. So far, no other studies have demonstrated the cytoplasmic expression of Sirt6; therefore, these findings seem novel. Please elaborate on the mechanism of nuclear–cytoplasmic shuttling of Sirt6.

Responses to Reviewers

We thank the reviewers for their thoughtful and constructive comments, which have guided the revision of this manuscript (Manuscript ID NCOMMS-16-18860-R2). The changes in the manuscript have been highlighted in yellow for easy identification. In addition, we also revised our manuscript based on the format requirement of Nature Communications. The concerns have been addressed as follows:

To Reviewer #1:

1. The paper is much improved, the conclusions clear. The new finding that Sirtuin6 mitigates podocyte uPAR expression is convincing that the author's mechanism is relevant to human glomerular disease. Given the importance of this observation, I would ask for this finding to be briefly mentioned in the abstract.

Answer: Thank you for your review and positive comments! As suggested, we highlighted the role of SIRT6 on the regulation of podocyte uPAR expression in the abstract (total abstract limit words: 150) in this revision. **The related modification was found in page 2 (abstract section).**

To Reviewer #3:

1. In Figure 1c, immunofluorescence has revealed that positive areas of immunostaining of Sirt6 were co-localized with synaptopodin. This seems inconsistent with the authors' reply to Major Comment 1 raised by Reviewer 3. The authors have written that Sirt6 was localized in the nucleus of the podocytes. However, synaptopodin was localized in the cytoplasm and not in the nucleus. This result has led us to infer that Sirt6 was also localized in the cytoplasm to some extent as shown in other types of cells. Please present the necessary immunofluorescence data that can confirm the authors' answer to our comment.

Answer: We really appreciate your careful review and make our results more reliable. Firstly, we redid immunofluorescence staining to detect the localization of SIRT6 and images were obtained by a higher resolution confocal microscope (ZEISS, Germany). As shown in the following figures, SIRT6 was mainly localized in the nucleus of the podocytes. By comparison, the inconsistency between two trials may attribute to the difference of accuracy of instruments and unspecific staining.

The new immunofluorescence data are much better to show that SIRT6 is mainly located in the nucleus of podocytes and the synaptopodin staining is mainly around the nucleus. **Representative podocytes were marked by arrows in the revised Figure 1c**, although we cannot completely exclude unspecific staining to some extent.

Meanwhile, we further stained Wilms' tumor protein-1 (WT-1), one nuclear protein which is one of the major markers for podocytes. It was found that WT-1 and SIRT6 mainly colocalized in the nuclei by immunofluorescence analysis. Considering that we identified not only for nuclear SIRT6 in podocytes, but also for other related proteins which are distributed in cytoplasm or membrane, therefore, synaptopodin is still a good candidate and used for podocyte markers in this manuscript. Therefore, **the related results for WT-1 staining were added in Supplementary Figure 1 as additional evidence.**

2. In Figure 1a, the authors have presented the immunoblots of seven isoforms of Sirt. Left panels demonstrate the results showing the expression levels in the sham group and STZ treatment group. Right panels demonstrate the results of the sham group and ADR administration group. The intensities of the bands corresponding to Sirt2, Sirt6, and Sirt7 in the sham groups demonstrated in the left panels seem very different from those demonstrated in the right panels. Because the loading dose of proteins in western blots and antibodies used are the same between both the panels, the immunoblot results should also be similar. However, the intensities of the bands were totally different. Please repeat the immunoblots.

Answer: Thank you very much for your question! As suggested, we repeated the immunoblots of seven isoforms of SIRTs. In fact, the length of exposure to film or digital imager is one important factor for immunoblot intensity. Although loading the same amount, the different exposure time leads to different intensities of the bands as exemplified by the following figure (see original data for SIRT2). To avoid misunderstanding, we try to make the

same exposure time in different films. **The related results was revised in Figure 1.**

3. In Figure 1b, the authors have presented representative immunoblots of Sirt6 in upper panels and quantification of their intensities in lower graphs. However, the results seem incompatible. For example, the levels of Sirt6 in db/db are approximately 40% as compared to the levels in db/+. However, the upper figures manifest either no bands or very slight bands for Sirt6 in db/db. Please explain the inconsistency between the intensities of the bands in the images and their quantification.

Answer: Thank you for your question! In fact, we have quantified the levels of Sirt6 in db/db according to different separated gels for different mice. Therefore, the intensities of the bands in the representative image were not very agreed to their quantification. Some of them were shown as follows. As suggested, we rescanned and analyzed the data, and made it more accurately in this revision. **The representative images and summarized data were modified in Figure 1.**

4. In Figure 5a, the authors have presented immunoblots of Sirt6 in left panels and their quantification in right graph. The intensities of the bands for Sirt6 adenovirus vector seem several hundred times higher than those for control adenovirus vector. However, the right graph illustrates that the expression of Sirt6 in adenoviral vector transfection was about four times higher than the expression of Sirt6 transfected in control vector. Please analyze the data more carefully.

Answer: Thank you for your question! As similar as the question 3, we have quantified the levels of Sirt6 according to different samples in different gels. Therefore, the intensities of the bands in the representative image were not very agreed to their quantification. Some of them were shown as follows. As suggested, we rescanned and analyzed the data and made it more accurately in this revision. **The representative images and summarized data were modified in Figure 5.**

In addition, we also carefully check other results and rescanned them to get more accurate statistical analysis.

5. As a reply to Major Comment from Reviewer 3, the authors have explained that other types of renal cells, such as glomerular endothelial cells, mesangial cells, and proximal tubules, showed both nuclear and cytoplasmic Sirt6, which indicated the possibility of nuclear translocation of Sirt6. So far, no other studies have demonstrated the cytoplasmic expression of Sirt6; therefore, these findings seem novel. Please elaborate on the mechanism of nuclear - cytoplasmic shuttling of Sirt6.

Answer: Thank you for your question! This is a big challenging question that we put our efforts for the distribution of SIRT6 in different cells, although this is not our major issue in the present study.

The subcellular localization of SIRT6 have always been an issue of debate. In previous study, SIRT1, SIRT6 and SIRT7 are found predominantly in the nucleus; SIRT3, SIRT4 and SIRT5 reside in mitochondria; SIRT2 is primarily cytoplasmic (Eriko Michishita et al. Evolutionarily conserved and nonconserved cellular localizations and functions of human SIRT proteins. *Mol Biol Cell.* 2005; 16: 4623-4635). However, Masaya Tanno et al found SIRT1 was expressed in the cytoplasm in some neuron-like cells of the striatum, and in ependymal cells, where SIRT1 was expressed in both the cytoplasm and nucleus (Masaya Tanno et al. Nucleocytoplasmic shuttling of the NAD-dependent histone deacetylase SIRT1. *J Biol Chem.* 2007; 282(9): 6823-6832). In addition, full-length (FL) SIRT3 was observed in the nucleus, different from the processed short form of SIRT3 which is a well-established mitochondrial protein (Toshinori Iwahara et al. SIRT3 functions in the nucleus in the control of stress-related gene expression. *Mol Cell Biol.* 2012; 32:5022 - 5034). Therefore, sirtuins are distributed among multiple compartments of the cell, and their localization may be dynamic, depending on tissue/cell type and physiologic conditions. In the present study, our results showed that SIRT6 localized in the nucleus of human podocytes, SIRT6 was also found more or less in the cytoplasm of some other renal cells by our current antibody-based molecular assays, leading us to speculate that renal SIRT6 may shuttle between the nucleus and cytoplasm, depending on cell type or environmental stimuli. Three possible reasons may explain the results, which need to be further investigated or demonstrated:

1. Ruth I. Tennen et al have found that the C-terminal extension (CTE) of SIRT6 contributes to proper nuclear localization, although the exact nuclear localization signal motif located in C-terminal extension (CTE) and the sequence of binding events has not been established for SIRT6 (Ruth I. Tennen et al. Functional dissection of SIRT6: identification of domains that regulate histone deacetylase activity and chromatin localization. *Mech Ageing Dev.* 2010; 131(3): 185–192). **According to the studies of other molecules, it is possible that some cofactors or proteins may have effect on the SIRT6 nuclear localization.** For example, a soluble fragment of the C-cadherin cytoplasmic domain is sufficient to inhibit the nuclear import of β -catenin, which indicates that the nuclear pore complex (NPC) contact surface of β -catenin might be masked (Suh E. K. et al. Translocation of β -catenin into the nucleus independent of interactions with FG-rich nucleoporins. *Exp. Cell Res.* 2003; 290: 447–456); SARA(SMAD adaptor for receptor activation), a FYVE domain-containing protein can bind to Smad2, and competes with the binding of FG-repeat containing nucleoporins, thereby blocking nuclear import (Xu L et al, Smad2

nucleocytoplasmic shuttling by nucleoporins CAN/Nup214 and Nup153 feeds TGF beta signaling complexes in the cytoplasm and nucleus. Mol. Cell; 2002; 10: 271–282). Proteins are translated in the cytoplasm, but many need to access the nucleus to perform their functions. Therefore, we speculate that there may be some factors existed to inhibit the nuclear import of SIRT6.

2. Understanding how these nuclear proteins are transported through the nuclear envelope and how the import processes are regulated is an important aspect of understanding cell function. The structural basis of the principal nuclear import pathways involve transport factors that are members of the β -karyopherin (β -Kap) family (e.g., importin- β), which can bind cargo directly or through adaptor proteins (e.g., importin- α). Post-translational modifications of these transport factors or adaptor proteins result in the outcome of protein import into the nucleus. It is emerging that different β -Kap family members recognize different classes of cargoes, and moreover, tissue-specific expression of family members may differentially localize cargoes within different cell types (Mary Christie et al. Structural biology and regulation of protein import into the nucleus. J Mol Biol, 2016; 428: 2060–2090). In addition, importin- α variants can display preferences for specific nuclear localization sequence (NLS), which may be important for development and tissue-specific roles (Mary Christie et al. Structural biology and regulation of protein import into the nucleus. J Mol Biol, 2016; 428: 2060–2090). As a result, **we speculate the difference of importin- β or importin- α depending on cell type or physiologic condition may influence on the distribution of SIRT6.**
3. Normally, nucleoprotein are translocated into nuclei by the formation of an intact and functional nuclear import complex which consists of cargos, transport factors and adaptor proteins. During this process, in some cell lines at different situation, there might be some proportion of SIRT6 in cytoplasm which have not formed nuclear import complex before translocation into nuclear. In this study, we used anti-SIRT6 antibody obtained from Abcam (cat No. AB62739; Lot No.GR285974) corresponding to N terminal amino acids 19-33 of Human SIRT6. The N terminal of SIRT6 is critical for catalytic activity and the C-terminal extension (CTE) of SIRT6 contributes to proper nuclear localization as we mentioned before (Ruth I. Tennen et al. Functional dissection of SIRT6: identification of domains that regulate histone deacetylase activity and chromatin localization. Mech Ageing Dev. 2010; 131(3): 185–192). Therefore, questions keep in our mind: 1. Is it possible that the SIRT6 antibody may bind to cytoplasm SIRT6 which have not formed intact and functional nuclear import complex? 2. The epitope of different antibodies is also a potential factor?

In summary, the distribution and the nucleocytoplasmic shuttling of SIRT6 is very complex. Although this is not our major issue in the present study, we will put more efforts to elucidate related mechanisms as an independent project based on our further observation.

REVIEWERS' COMMENTS:**Reviewer #3 (Remarks to the Author):**

Basically, I have confirmed the authors' responses and many additional results. However, I still feel that Sirt6 regulates multiple downstream molecules in podocytes as compared to the the specific tartets in other organs as previous reports demonstrated. In the discussion section, please consider to add the explanation about it.

Responses to Reviewers

We thank the reviewers for their thoughtful and positive comments, which have guided the revision of this manuscript (Manuscript ID NCOMMS-16-18860-R3). The changes in the manuscript have been highlighted in yellow for easy identification. In addition, we also revised our manuscript based on the format requirement of Nature Communications. The concerns have been addressed as follows:

To Reviewer #3:

1. Basically, I have confirmed the authors' responses and many additional results. However, I still feel that Sirt6 regulates multiple downstream molecules in podocytes as compared to the specific targets in other organs as previous reports demonstrated. In the discussion section, please consider to add the explanation about it.

Answer: Thanks for the reviewer's suggestion. In this study, we cannot exclude other downstream molecules of Sirt6 may also contribute to the regulation of podocyte function. For instance, Wnt/ β -catenin signaling is an evolutionarily conserved developmental signaling cascade that exhibits a pivotal function in promoting podocyte dysfunction and albuminuria in proteinuric kidney diseases (Dai C, *et al.* Wnt^{-/-}-Catenin Signaling Promotes Podocyte Dysfunction and Albuminuria. **J Am Soc Nephrol**, 2009, 20: 1997–2008). Considering that studies have demonstrated that SIRT6 controls hematopoietic stem cell homeostasis through epigenetic regulation of Wnt signaling (Hu Wang, *et al.* SIRT6 Controls Hematopoietic Stem Cell Homeostasis through Epigenetic Regulation of Wnt Signaling. **Cell Stem Cell**, 2016, 18: 495–507), it is possible that Sirt6-mediated Wnt signaling is also associated with the regulation of podocyte function.

In addition, other downstream molecules of Sirt6 such as NF- κ B p65 (Zhang N, *et al.* Calorie restriction-induced SIRT6 activation delays aging by suppressing NF- κ B signaling. **Cell Cycle**, 2016; 5(7):1009-18) and AMPK (Elhanati S, *et al.* Multiple Regulatory Layers of SREBP1/2 by SIRT6. **Cell Rep**, 2013, 4(5):905-12) may also contribute to the regulation of podocyte function. Therefore, it is necessary to further clarify the role of Wnt signaling as well as other potential downstream targets regulated by Sirt6 in podocytes.

As suggested, we briefly discuss this issue in this revision. **The related modification was found in page 14.**